# A Simulation-based Framework for Robust Federated Learning to Training-time Attacks

## Abstract

Well-known robust aggregation schemes in federated learning (FL) are shown to be vulnerable to an informed adversary who can tailor training-time attacks (Fang et al., 2020; Xie et al., 2020). We frame robust distributed learning problem as a game between a server and an adversary that is able to optimize strong training-time attacks. We introduce RobustTailor, a simulation-based framework that *prevents the adversary from being omniscient*. The simulated game we propose enjoys theoretical guarantees through a regret analysis. RobustTailor improves robustness to training-time attacks significantly while preserving almost the same privacy guarantees as standard robust aggregation schemes in FL. Empirical results under challenging attacks show that RobustTailor performs similar to an *upper bound* with perfect knowledge of honest clients.

## 1 Introduction

In *federated learning* (FL), a global/personalized model is learnt from data distributed on multiple clients without sharing data (McMahan et al., 2017; Kairouz et al., 2021). Clients compute their (stochastic) gradients using their own local data and send them to a central server for aggregating and updating a model. While FL offers improvements in terms of privacy, it creates additional challenges in terms of robustness. Clients are often prone to the bias in the *stochastic gradient* updates, which comes not only from poor sampling or data noise but also from *malicious attacks of Byzantine clients* who may send arbitrary messages to the server instead of correct gradients (Guerraoui et al., 2018). Therefore, in FL, it is essential to guarantee some level of robustness to Byzantine clients that might be compromised by an adversary.

Compromised clients are vulnerable to data/model poisoning and tailored attacks (Fang et al., 2020). Byzantine-resilience is typically achieved by robust gradient aggregation schemes *e.g.,* Krum (Blanchard et al., 2017), Comed (Yin et al., 2018), and trimmedmean (Yin et al., 2018). These aggregators are resilient against attacks that are designed in advance. However, such robustness is insufficient in practice since a powerful adversary could learn the aggregation rule and tailor its training-time attack. It has been shown that well-known Byzantine-resilient gradient aggregation schemes are susceptible to an informed adversary that can tailor the attacks (Fang et al., 2020). Specifically, Fang et al. (2020) and Xie et al. (2020) proposed efficient and nearly optimal training-time attacks that circumvent Krum, Comed, and trimmedmean. A tailored attack is designed with a prior knowledge of the robust aggregation rule used by the server, such that the attacker has a provable way to corrupt the training process. Given the information leverage of the adversary, it is a significant challenge to establish successful defense mechanisms against such tailored attacks.

In this paper, we formulate robust distributed learning problem against training-time attacks as a game between a server and an adversary. To prevent the adversary from being omniscient, we propose to follow a *mixed strategy* using the existing robust aggregation rules. In real-world settings, both server and adversary have a number of aggregation rules and attack programs. How to utilize these aggregators efficiently and guarantee robustness is a challenging task. We address scenarios where neither the specific attack method is known in advance by the aggregator nor the exact aggregation rule used in each iteration is known in advance by the adversary, while the adversary and the server know the set of server's aggregation rules and the set of attack programs, respectively.[1]

---

[1]While this assumption is essential to frame our game, we provide experimental results on challenging settings where the server does not know the set of attack programs in Section 5.

Due to information asymmetry between the server and the adversary, we assume every client donates *a small amount of honest data* to the server as the price to achieve some level of security more proactively and efficiently. Providing such public dataset to achieve robustness is a common assumption in FL (Fang & Ye, 2022; Huang et al., 2022; Kairouz et al., 2021; Yoshida et al., 2020; Zhao et al., 2018; Fang et al., 2020; Xie et al., 2020; Cao & Lai, 2019; Chang et al., 2019; Cao et al., 2020). We propose RobustTailor, a scheme based on simulating aggregation rules under different attacks. With minimal privacy leakage, RobustTailor realizes high resilience to training-time attacks. RobustTailor maintains stable performance under a challenging mixed attack, a strategy we propose for the adversary to simulate and design a successful attack when a smart server uses a mixed strategy to make the problem of attack design computationally harder. We emphasize that any deterministic Byzantine-resilient algorithm can be added in server's aggregation pool. Similarly, any attack can be used in the set of adversary's attack programs.

## 1.1 SUMMARY OF CONTRIBUTIONS

- We frame robust distributed learning problem as *a game* between a server and an adversary that tailors training-time attacks.
- We propose a simulation-based framework RobustTailor to improve robustness by preventing the adversary from being omniscient.
- The simulated game we propose enjoys theoretical guarantees through a regret analysis.
- Empirical studies validate our theory and show that RobustTailor *preforms similar to an upper bound* with perfect knowledge of all honest clients over the course of training. Even under a challenging mixed attack strategy, RobustTailor outperforms the robust baselines in terms of robustness and accuracy.

## 1.2 RELATED WORK

In this section, we provide a summary of related work. See Appendix A for complete related work.

**Training-time attacks in FL.** Federated learning (FL) usually suffers from training-time attacks (Biggio et al., 2012; Bhagoji et al., 2019; Sun et al., 2019; Bagdasaryan et al., 2020) because the server trains the model across various unreliable clients with private datasets. A strong adversary can potentially participate in every training round and adapt its attacks to an updated model. In *model update poisoning*, a class of training-time attacks, an adversary controls some clients and directly manipulates their outputs aiming to bias the global model towards opposite direction of honest training (Kairouz et al., 2021). If Byzantine clients have access to the updates of honest clients, they can tailor their attacks and make them difficult to detect (Fang et al., 2020; Xie et al., 2020).

**Robust aggregation and Byzantine resilience.** To improve robustness under general Byzantine clients, a number of robust aggregation schemes have been proposed, which are mainly inspired by robust statistics such as median-based aggregators (Yin et al., 2018; Chen et al., 2017), Krum (Blanchard et al., 2017), trimmed mean (Yin et al., 2018). Moreover, Fang et al. (2020); Xie et al. (2020); Cao & Lai (2019); Cao et al. (2020) propose server-side verification methods using auxiliary data. Karimireddy et al. (2021) and Alistarh et al. (2018) propose history-aided aggregators. Ramezani-Kebrya et al. (2022) propose a framework based on randomization of multiple aggregation rules. However, none of them selects a proper aggregation rule proactively during training as our framework RobustTailor, and all of them can be used in RobustTailor while we mainly focus on statistical-based aggregators in this paper. Although past work has shown that these aggregators can defend successfully under specific conditions (Blanchard et al., 2017; Chen et al., 2017; Su & Vaidya, 2016), Fang et al. (2020) and Xie et al. (2020) argue that Byzantine-resilient aggregators can fail when an informed adversary tailor a careful attack and Gouissem et al. (2022) proves that such aggregation rules are vulnerable. Therefore, developing a robust and efficient algorithm under such strong tailored attacks is essential to improve security of FL, which is the goal of this paper.

**Game theory in FL.** Online convex optimization (OCO) framework (Zinkevich, 2003) is widely influential in the learning community (Hazan et al., 2016; Shalev-Shwartz et al., 2012), and bandit convex optimization (BCO) as an extension of OCO was proposed by Awerbuch & Kleinberg (2008) for decision making with limited feedback. Bandit paradigms paralleling FL framework are proposed by Shi & Shen (2021) and its extension under Byzantine attacks is proposed by Demirel et al. (2022). However, they account for uncertainties from both arm and client sampling rather than robust

aggregation in this paper. In this paper, we frame robust distributed learning problem as a game and consider the *bandit feedback model*.

## 2 PROBLEM SETTING

Under a synchronous setting in FL, clients compute their updates on their own local data and then aggregate from all peers to update model parameters. Consider a general distributed system consisting of a parameter server and $n$ clients (Chen et al., 2017; Abadi et al., 2016; Li et al., 2014). Suppose that $f$ Byzantine clients are controlled by an adversary and behave arbitrarily. Let $\mathbf{x} \in \mathbb{R}^d$ denote a machine learning model, *e.g.,* it represents all weights and biases of a neural network. We consider minimizing an overall empirical risk of multiple clients, which can be formulated as finite-sum problem:

$$\min_{\mathbf{x} \in \mathbb{R}^d} F(\mathbf{x}) = \frac{1}{n} \sum_{i=1}^{n} F_i(\mathbf{x}) \tag{FL}$$

where $F_i : \mathbb{R}^d \to \mathbb{R}$ denotes the training error (empirical risk) of $\mathbf{x}$ on the local data of client $i$.

At iteration $t$, *honest* clients compute and send *honest* stochastic gradients $\mathbf{g}_i(\mathbf{x}_t) = \nabla F_i(\mathbf{x}_t)$ for $i \in [n - f]$ while *Byzantine* clients, controlled by an informed adversary, output *attacks* $\mathbf{b}_j \in \mathbb{R}^d$ for $j \in [f]$. The server receives all $n$ updates and aggregates them following a particular *robust* aggregation rule, which outputs an aggregated and updated model $\mathbf{x}_{t+1} \in \mathbb{R}^d$. Finally, the server broadcasts $\mathbf{x}_{t+1}$ to all clients.

### 2.1 GAME CONSTRUCTION

We frame this distributed learning problem under training-time attack as a game played by the adversary and the server. The informed adversary and training-time attacks are described in Section 2.1.1. The details of aggregators for the server are provided in Section 2.1.2. Though our formulation seems natural and intuitive, to the best of our knowledge, our work is the *first work* that frames robust learning problem under training-time tailored attacks as a game. The adversary aims at corrupting training while the server aims at learning an effective model, which achieves a satisfactory overall empirical risk over honest clients.

#### 2.1.1 INFORMED ADVERSARY WITH ATTACKS

The adversary controls $f$ out of $n$ clients where these Byzantine clients collude aiming at disturbing the entire training process by sending training-time attacks (Biggio et al., 2012; Bhagoji et al., 2019; Sun et al., 2019; Bagdasaryan et al., 2020). We assume $n \geq 2f + 1$ which is a common assumption in the literature (Guerraoui et al., 2018; Blanchard et al., 2017; Alistarh et al., 2018; Rajput et al., 2019; Karimireddy et al., 2022); otherwise the adversary will be able to provably control the optimization trajectory and set the global model arbitrarily (Lamport et al., 1982).

An *informed adversary* controls the outputs of those compromised clients, *e.g.,* their gradients throughout the course of training. Moreover, the informed adversary has *full knowledge* of the outputs of $n - f$ honest clients across the course of training. Having access to the gradients of honest nodes, the adversary can compute the global aggregated gradient of an omniscient aggregation rule, which is the empirical mean of *all honest updates without an attack*:

$$\mathbf{g}^* = \frac{1}{n - f} \sum_{i=1}^{n-f} \mathbf{g}_i. \tag{1}$$

When an adversary knows a particular server's aggregation rule, it is able to design tailored attacks using $n - f$ honest gradients (Fang et al., 2020).

**Definition 1** (Attack algorithm). *Let $\{\mathbf{g}_1, \ldots, \mathbf{g}_{n-f}\}$ denote the set of honest updates computed by $n - f$ honest clients. The adversary designs $f$ Byzantine updates using an AT algorithm:*

$$\{\mathbf{b}_{n-f+1}, \ldots, \mathbf{b}_n\} := \mathrm{AT}(\mathbf{g}_1, \ldots, \mathbf{g}_{n-f}, \mathcal{A}). \tag{2}$$

*where $\mathcal{A}$ denotes the set of aggregators formally defined in Section 2.1.2.*

It is shown that several tailored attacks can be designed efficiently and provably fail well-known aggregation rules with a specific structure *e.g.,* Krum, Comed, and Krum + resampling (Fang et al., 2020; Xie et al., 2020; Ramezani-Kebrya et al., 2022). Suppose that the adversary has a set of $S$ computationally tractable programs to design tailored attacks:

$$\mathcal{F} = \{\mathrm{AT}_1, \mathrm{AT}_2, \ldots, \mathrm{AT}_S\}. \tag{3}$$

### 2.1.2 SERVER WITH AGGREGATORS

The server aims at learning an effective model, which achieves a satisfactory overall empirical risk over honest clients comparable to that *under no attack*. To update the global model, the server aggregates all gradients sent by clients at each iteration.

**Definition 2** (Aggregation rule). *Let $\mathbf{g}'_j \in \mathbb{R}^d$ denote an update received from client $j$, which can be either an honest or compromised client for $j \in [n]$. The server aggregates all updates from $n$ clients and outputs a global update $\mathbf{g} \in \mathbb{R}^d$ using an aggregation rule $\mathrm{AG}$:*

$$\mathbf{g} = \mathrm{AG}(\mathbf{g}'_1, \ldots, \mathbf{g}'_n, \mathcal{F}). \tag{4}$$

*where $\mathcal{F}$ denotes the set of attacks defined in Section 2.1.1.*

We assume that the server knows the number of compromised clients $f$ or an upper bound on $f$, which is a common assumption in robust learning (Guerraoui et al., 2018; Blanchard et al., 2017; Alistarh et al., 2018; Rajput et al., 2019; Karimireddy et al., 2022).

However, the server does not know the specific Byzantine clients among $n$ clients in this distributed system such that the server cannot compute $\mathbf{g}^*$ in Eq. (1) directly. To learn and establish some level of robustness against training-time attacks, several Byzantine-resilient aggregation rules have been proposed *e.g.,* Krum (Blanchard et al., 2017) and Comed (Yin et al., 2018). These methods inspired by robust statistics provide rigorous convergence guarantees under *specific settings* and have been shown to be vulnerable to tailored attacks (Fang et al., 2020; Xie et al., 2020). The set of $M$ aggregators used by the server is denoted by

$$\mathcal{A} = \{\mathrm{AG}_1, \mathrm{AG}_2, \ldots, \mathrm{AG}_M\}. \tag{5}$$

Note that the pool of aggregators $\mathcal{A}$ and the set of attacks $\mathcal{F}$ are known by both the server and the adversary, but the specific $\mathrm{AT}^t$ are $\mathrm{AG}^t$ chosen at iteration $t$ are unknown. Moreover, such powerful but *not omniscient* adversary does not have access to the random seed generator at the server. This is a mild and common assumption in cryptography, which requires a secure source of entropy to generate random numbers at the server (Ramezani-Kebrya et al., 2022).

To avoid trivial solutions, we assume each aggregation rule is robust (formal definition of robustness is provided in Appendix B) against a *subset of attack algorithms* in $\mathcal{F}$ while no aggregation rule is immune to all attack algorithms. Similarly, each attack program can *provably fail* one or more aggregation rules in $\mathcal{A}$ while no attack program can provably fail *all aggregation rules*.

## 2.2 PROBLEM FORMULATION

To evaluate the performance of an updated global model, *i.e.,* the output of AG in Eq. (4), we define a loss function, which measures the discrepancy of the output of AG and an omniscient model update *under no attack*.

**Definition 3** (Loss function). *The loss function associated with using aggregation rule $\mathrm{AG}$ under attack $\mathrm{AT}$ is defined as:*

$$
\begin{aligned}
\ell(\mathrm{AG}, \mathrm{AT}, \{\mathbf{g}'_i\}_{i=1}^n) &= \|\mathrm{AG}(\mathbf{g}_1, \ldots, \mathbf{g}_{n-f}, \mathrm{AT}(\mathbf{g}_1, \ldots, \mathbf{g}_{n-f}, \mathcal{F}), \mathcal{A}) - \mathbf{g}^*\| \\
&= \|\mathrm{AG}(\mathbf{g}_1, \ldots, \mathbf{g}_{n-f}, \mathbf{b}_{n-f+1}, \ldots, \mathbf{b}_n, \mathcal{A}) - \mathbf{g}^*\|.
\end{aligned}
\tag{6}
$$

*where $\mathbf{g}^*$ is the ideal model under no attack which is computed in Eq. (1).*

To train the global model, the server takes multiple rounds of stochastic gradient descent by aggregating the stochastic gradients from clients. However, some gradients might be corrupt at each round, which are sent by compromised clients controlled by the adversary. We frame this robust distributed

learning scenario as a *game* between the adversary and the server. The server wants to minimize the loss defined in Definition 3, while the adversary aims to maximize it. This game as a *minimax optimization problem* is formulated as:

$$\min_{\mathrm{AG} \in \mathcal{A}} \max_{\mathrm{AT} \in \mathcal{F}} \ell(\mathrm{AG}, \mathrm{AT}, \{\mathbf{g}'_i\}_{i=1}^n). \tag{MinMax}$$

The entire process of model aggregation with $T$ rounds is shown in Algorithm 1. Note that $\mathbb{E}_{\mathcal{G}}$ denotes the expectation with respect to randomness due to stochastic gradients. Ideally, the game in MinMax can reach a Nash equilibrium (NE) (Nash, 1950). However, the loss is not computable for the server, since the server cannot distinguish honest gradients such that $\mathbf{g}^*$ is unknown for it. Therefore, we will *simulate* the game in the following Section 3.

---

**Algorithm 1** Update model from the perspective of server

---

**Input:** Initial weight vector $\mathbf{x}_0$, learning rate $\eta_t$, iteration rounds $T$, number of clients $n$, set of aggregation rules $\mathcal{A}$, set of attack algorithms $\mathcal{F}$, and public dataset.

1 **for** $t = 1$ *to* $T$ **do**
2     Server sends $\mathbf{x}_t$ to all clients.
3     **for** $i = 1$ *to* $n - f$ **do**
4         | Honest client $i$ computes local gradient $\mathbf{g}_i(\mathbf{x}_t) = \nabla F_i(\mathbf{x}_t)$.
5     Compromised clients send attacks $\mathrm{AT}^t(\{\mathbf{g}_i\}_{i=1}^{n-f}, \mathcal{A})$.
6     Sever receives gradients from all clients $\{\mathbf{g}'_i\}_{i=1}^n$.
7     Server chooses $\mathrm{AG}^t$ by solving

$$\min_{\mathrm{AG}^t \in \mathcal{A}} \max_{\mathrm{AT}^t \in \mathcal{F}} \mathbb{E}_{\mathcal{G}}[\ell(\mathrm{AG}^t, \mathrm{AT}^t, \{\mathbf{g}'_i\}_{i=1}^n)].$$

8     Server updates the model $\mathbf{x}_{t+1} = \mathbf{x}_t - \eta_t \mathrm{AG}^t(\{\mathbf{g}'_i\}_{i=1}^n, \mathcal{F})$.

---

## 3 ROBUST AGGREGATION

Because MinMax cannot be solved during the process of updating the model, we propose to *simulate* it instead and obtain an optimized aggregator for model updates. As mentioned in Section 2, the informed adversary has *an advantage* over the server since it can perfectly estimate $\mathbf{g}^*$ in Eq. (1) while the server does not have such knowledge and cannot identify honest clients a priori. We assume that each client donates a *small amount* of data as a *public dataset* to the server to achieve some level of security by controlling the *information* gap between the server and adversary. Let $\widetilde{\mathbf{g}}$ denote the update computed at the server using the public dataset, which is a rough estimate of $\mathbf{g}^*$.

**Remark 1.** *The server may obtain such public dataset from other sources not necessarily the current clients. It is sufficient as long as the collected public dataset represents the clients' data distribution by some extent. To guarantee convergence, we only require that the update from the public dataset is a **rough estimate** of the ideal $\mathbf{g}^*$. In particular, the quality of such estimate directly impacts the convergence of our proposed algorithm (see Section 4 for details). As the quality of such estimate improves, the convergence of the global model to an effective model improves. The existence of such public dataset is a valid and common assumption in FL (Fang & Ye, 2022; Huang et al., 2022; Kairouz et al., 2021; Yoshida et al., 2020; Chang et al., 2019; Zhao et al., 2018).*

For the simulation, the server generates the simulated gradients $\{\widetilde{\mathbf{g}}_i\}_{i=1}^{n-f}$ based on the public dataset. The loss function in the simulated game becomes

$$\widetilde{\ell}(\mathrm{AG}, \mathrm{AT}, \{\widetilde{\mathbf{g}}_i\}_{i=1}^{n-f}) = ||\mathrm{AG}(\widetilde{\mathbf{g}}_1, \dots, \widetilde{\mathbf{g}}_{n-f}, \mathrm{AT}((\widetilde{\mathbf{g}}_1, \dots, \widetilde{\mathbf{g}}_{n-f}, \mathcal{A}), \mathcal{F}) - \frac{1}{n-f}\sum_{i=1}^{n-f}\widetilde{\mathbf{g}}_i|| \tag{7}$$

Let $\mathcal{L} \in \mathbb{R}_+^{M \times S}$ denote the loss of $M$ aggregators corresponding to $S$ attacks, and $\mathcal{L}(\mathrm{AG}_i, \mathrm{AT}_j)$ represents the loss associated with aggregation rule $i$ in $\mathcal{A}$ under attack $j$ in $\mathcal{F}$ in the simulation. After the adversary has committed to a probability distribution $\mathbf{q}$ over $S$ attack algorithms, the server chooses a probability distribution $\mathbf{p}$ over $M$ aggregation rules. Then, the server incurs the loss $\widetilde{\ell}(\mathbf{p}, \mathbf{q}) = \mathbf{p}\mathbb{E}_{\mathcal{G}}[\mathcal{L}]\mathbf{q}^\top$. We will solve Sim-MinMax below instead of MinMax.

$$\min_{\mathbf{p} \in \Delta_M} \max_{\mathbf{q} \in \Delta_S} \mathbf{p}\mathbb{E}_{\mathcal{G}}[\mathcal{L}]\mathbf{q}^\top. \tag{Sim-MinMax}$$

where $\Delta_M$ and $\Delta_S$ denote the probability simplex in $[M]$ and $[S]$, respectively.

In practice, it is computationally expensive to compute $\mathcal{L}^{M \times S}$ and meanwhile there is noise due to the gradients. Therefore, we consider *bandit feedback model* with limited feedback, in which the server and adversary can only observe the loss through exploration. To solve Sim-MinMax in the bandit feedback model, one player could implement the well-known *Exponential-weight Algorithm for Exploration and Exploitation (Exp3)* (Seldin et al., 2013) whose detailed description is deferred to Appendix C.

Due to two players (the server and the adversary) in our model, we propose an algorithm which simultaneously executes Exp3. We term our proposed *robust aggregation scheme* as RobustTailor, which outputs an optimized AG at each iteration. The specific algorithm from perspective of the server is shown in Algorithm 2. Using the public dataset, the server generates $n - f$ noisy stochastic gradients $\widetilde{\mathbf{g}}_i = \nabla \widetilde{F}_i(\mathbf{x}_t)$ for $i \in [n - f]$ at iteration $t$. After $K$ rounds of simulation on $\{\widetilde{\mathbf{g}}_i\}_{i=1}^{n-f}$, the server obtains a final probability distribution $\mathbf{p}$ and selects an aggregation rule by sampling from $\mathbf{p}$. The steps for our robust training procedure are summarized in Algorithm 3. Note that Appendix H demonstrates both theoretical analysis and empirical results of RobustTailor's computation complexity.

---

**Algorithm 2** RobustTailor

---

**Input:** Updating rates $\lambda_1, \lambda_2, \tilde{\lambda}_1$ and $\tilde{\lambda}_2$, simulation rounds $K$, simulated gradients $\{\widetilde{\mathbf{g}}_i\}_{i=1}^{n-f}$, $\mathcal{A}$, $\mathcal{F}$.
1  Initialize weight vector $w^0(i) = 1$ for $i \in [M]$ and $v^0(j) = 1$ for $j \in [S]$.
2  **for** $k = 1$ **to** $K$ **do**
3      Set $\mathbf{p}^k(\widehat{\mathrm{AG}}_i) = (1 - \lambda_1) \frac{w^k(i)}{\sum_{i=1}^{M} w^k(i)} + \lambda_1 \frac{1}{M}$ for $i \in [M]$.
4      Set $\mathbf{q}^k(\widehat{\mathrm{AT}}_j) = (1 - \lambda_2) \frac{v^k(j)}{\sum_{j=1}^{S} v^k(j)} + \lambda_2 \frac{1}{S}$ for $j \in [S]$.
5      Sample $\mathrm{AG}^k \sim \mathbf{p}^k$ and $\mathrm{AT}^k \sim \mathbf{q}^k$ respectively.
6      Estimate the loss $\ell^k = \widetilde{\ell}(\mathrm{AG}^k, \mathrm{AT}^k, \{\widetilde{\mathbf{g}}_i\}_{i=1}^{n-f})$.
7      Set $\hat{\ell}_1^k(i) = \frac{\mathbb{I}\{\widehat{\mathrm{AG}}_i = \mathrm{AG}^k\}}{\mathbf{p}^k(\widehat{\mathrm{AG}}_i)} \ell^k$, $\quad w^{k+1}(i) = w^k(i) \exp(-\tilde{\lambda}_1 \hat{\ell}_1^k(i)/M)$ for $i \in [M]$.
8      Set $\hat{\ell}_2^k(j) = \frac{\mathbb{I}\{\widehat{\mathrm{AT}}_j = \mathrm{AT}^k\}}{\mathbf{q}^k(\widehat{\mathrm{AT}}_j)} \ell^k$, $\quad v^{k+1}(j) = v^k(j) \exp(\tilde{\lambda}_2 \hat{\ell}_2^k(j)/S)$ for $j \in [S]$.
9  Set $\mathbf{p}_i = \frac{\sum_{k=1}^{K} \mathbf{p}^k(\widehat{\mathrm{AG}}_i)}{K}$ for $i \in [M]$.
10  Sample $\mathrm{AG} \sim \mathbf{p}$.
**Output:** AG.

---

**Algorithm 3** Server's aggregation

---

**Input:** Learning rate $\eta_t$, $n$ clients, $f$ compromised clients, iteration rounds $T$, $\mathcal{A}$ and $\mathcal{F}$
1  Initialize model $\mathbf{x}_0$.
2  **for** $t = 1$ **to** $T$ **do**
3      Send $\mathbf{x}_t$ to all clients.
4      Receive gradients from all clients $\{\mathbf{g}_i'\}_{i=1}^{n}$.
5      Calculate simulated gradients $\{\widetilde{\mathbf{g}}_i\}_{i=1}^{n-f}$.
6      Call Algorithm 2 to aggregate $\mathrm{AG}^t = \mathrm{RobustTailor}(\{\widetilde{\mathbf{g}}_i\}_{i=1}^{n-f}, \mathcal{A}, \mathcal{F})$.
7      Update the global model by $\mathbf{x}_{t+1} = \mathbf{x}_t - \eta_t \mathrm{AG}^t(\{\mathbf{g}_i'\}_{i=1}^{n}, \mathcal{F})$.

---

The adversary can also perform simulation to *optimize* its attack at each iteration. The main differences for an adversarial simulation compared to RobustTailor include: 1) the adversary can use perfect honest stochastic gradients $\{\mathbf{g}_i\}_{i=1}^{n-f}$ instead of noisy estimates; 2) the probability output is $\mathbf{q}$ which is calculated by the weight vector of attacks $v(j)$ for $j \in [S]$. The details of the adversarial simulation are provided in Appendix D.

## 4 THEORETICAL GUARANTEES

To show convergence of the inner optimization in Algorithm 2, we first show how to turn two simultaneously played no-regret algorithms for a minimax problem into convergence to a Nash equilibrium (NE). To make the optimization problem shown in Algorithm 2 more general, we define a new loss

function $L : [M] \times [S] \to \mathbb{R}_+$. Consider simultaneously running two algorithms on the objective $L$, such that their respective expected regret is upper bounded by some quantities $\mathcal{R}_K^{\mathrm{i}}$ and $\mathcal{R}_K^{\mathrm{j}}$, i.e.,

$$\mathbb{E}\left[\sum_{k=1}^K L(i_k, j_k) - \sum_{k=1}^K L(i, j_k)\right] \le \mathcal{R}_K^{\mathrm{i}}, \quad \mathbb{E}\left[\sum_{k=1}^K L(i_k, j) - \sum_{k=1}^K L(i_k, j_k)\right] \le \mathcal{R}_K^{\mathrm{j}}, \quad (8)$$

for any $i \in [M]$ and $j \in [S]$ where the expectation is taken over the randomness of the algorithms.

**Lemma 1** (Folklore). *Assume we run two algorithms simultaneously with regret as in* (8) *to obtain* $\{(i_k, j_k)\}_{k=1}^K$. *By playing $\bar{i}$ uniformly sampled from $\{i_k\}_{k=1}^K$, we can guarantee*

$$\mathbb{E}_{\bar{i}}\left[L(\bar{i}, j)\right] \le \mathbb{E}_{i^\star \sim \mathbf{p}^\star, j^\star \sim \mathbf{q}^\star}\left[L(i^\star, j^\star)\right] + \frac{1}{K}(\mathcal{R}_K^{\mathrm{i}} + \mathcal{R}_K^{\mathrm{j}}), \quad (9)$$

*for any $j \in [S]$ where $(\mathbf{p}^\star, \mathbf{q}^\star)$ is a Nash equilibrium of $\mathbb{E}_{i^\star \sim \mathbf{p}^\star, j^\star \sim \mathbf{q}^\star}\left[L(i^\star, j^\star)\right]$.*

This kind of result is well-known in the literature (see for instance Dughmi et al. (2017, Cor. 4)). When the algorithms have sublinear regrets, we refer to them as *no-regret* algorithms. This condition ensures that the error term in (9) vanishes as $K \to \infty$. Exp3 (Auer et al., 2002), employed by both the attacker and aggregator in Algorithm 2, enjoys such a no-regret property.

**Lemma 2** (Hazan et al. 2016, Lemma 6.3). *Let $K$ be the horizon, $N$ be the number of actions, and $L_k : [N] \to \mathbb{R}_+$ be non-negative losses for all $k$. Then Exp3 with stepsize $\lambda = \sqrt{\frac{\log N}{KN}}$ enjoys the following regret bound,*

$$\mathbb{E}\left[\sum_{k=1}^K L_k(i_k) - \sum_{k=1}^K L_k(i)\right] \le 2\sqrt{KN \log N}, \quad (10)$$

*for any $i \in [N]$, where the expectation is taken over the randomness of the algorithm.*

Note that any two simultaneously played no-regret algorithms for a minimax problem can be turned into convergence to a NE following Lemmas 1 and 2. We obtain guarantees for the aggregation rule returned from Algorithm 2 as a direct consequence of Lemmas 1 and 2. Considering a specific situation in Algorithm 2, $L$ is replaced by the simulation loss $\widetilde{\ell}$ shown in Eq. (7).

**Lemma 3.** *Let $\widetilde{\ell}$ be the simulation loss in Eq.* (7). *Sample $\mathrm{AG} \sim \mathbf{p}$ as defined in Algorithm 2 with $\tilde{\lambda}_1 = \sqrt{\frac{\log M}{KM}}$ and $\tilde{\lambda}_2 = \sqrt{\frac{\log S}{KS}}$. Then the loss is bounded in expectation for any attack $\mathrm{ATK} \in \mathcal{F}$ as,*

$$\mathbb{E}_{\mathrm{AG}, \mathcal{G}}\left[\widetilde{\ell}\left(\mathrm{AG}, \mathrm{AT}, \{\mathbf{g}_i'\}_{i=1}^n\right)\right] \le \mathbf{p}^\star \mathbb{E}_{\mathcal{G}}[\mathcal{L}](\mathbf{q}^\star)^\top + 2\frac{\sqrt{M \log M} + \sqrt{S \log S}}{\sqrt{K}}, \quad (11)$$

*where $(\mathbf{p}^\star, \mathbf{q}^\star) \in \Delta_M \times \Delta_S$ is a Nash equilibrium of the zero-sum game with stochastic payoff matrix $\mathbb{E}_{\mathcal{G}}[\mathcal{L}]$ as defined in Sim-MinMax.*

Lemma 3 implies that the simulated loss approaches the NE value even under the worst-case attack. The proofs of Lemma 1 and Lemma 3 are provided in Appendix E and Appendix F respectively. Importantly, a sufficient condition for almost sure convergence for the *outer* loop is provided in Appendix B.

## 5 EXPERIMENTAL EVALUATION

In this section, we evaluate the resilience of RobustTailor against tailored attacks. To provide intuitive results and show benefits of simulation in terms of robustness, we first construct a simple pool of aggregators including only Krum (Blanchard et al., 2017) and Comed (Yin et al., 2018). Concerning the adversary's tailored attacks, we consider two different types of attacks, which can successfully ruin Krum and Comed, respectively. As described in (Fang et al., 2020; Xie et al., 2020), an adversary with $\epsilon$-*reverse attack* computes the average of honest update, scales the average with a parameter $\epsilon$, and sends scaled updates to the server to induct the global model towards the inverse of the direction along the one without attacks. It is known that a small $\epsilon$ corrupts Krum, while a large one corrupts Comed (Fang et al., 2020; Xie et al., 2020).

We simulate training with a total of 12 clients, 2 of which are compromised by an informed adversary. We train a CNN model on MNIST (Lecun et al., 1998) under both iid and non-iid settings and meanwhile training CNN models on Fashion-MNIST (FMNIST) (Xiao et al., 2017) and CIFAR10 (Krizhevsky et al., 2009) is under iid setting. Note that the dataset is shuffled and equally partitioned among clients in the iid settings (Fig. 1). In addition, all honest clients donate 5% of their local training dataset to the server as a public dataset as specified in Remark 1, and the informed adversary has access to the gradients of honest clients. Note that the details of the model and training hyper-parameters are provided in Appendix G.1, and all experiments below without specific clarification have same setup with Fig. 1a, Fig. 1b, and Fig. 3.

**Single tailored attacks.** RobustTailor successfully decreases the capability of the adversary to launch tailored attacks. RobustTailor maintains stability in Fig. 1a when Krum fails catastrophically under a small $\epsilon$ attack. Fig. 1b shows that RobustTailor has much less fluctuations in terms of test accuracy compared to Comed when facing a large $\epsilon$ attack. In addition, on average, RobustTailor has 70.68% probability of choosing Comed under $\epsilon = 0.5$ attack while 65.49% probability of choosing Krum under $\epsilon = 100$ attack, which proves that the server successfully learns how to defend. Training on FMNIST shows consistent results as seen in Fig. 1c and Fig. 1d and further results on CIFAR10 are in Appendix G.2. Note that RobustTailor will outperform both Krum and Comed if there is a larger pool to select aggregators, which results are shown in Appendix G.2.

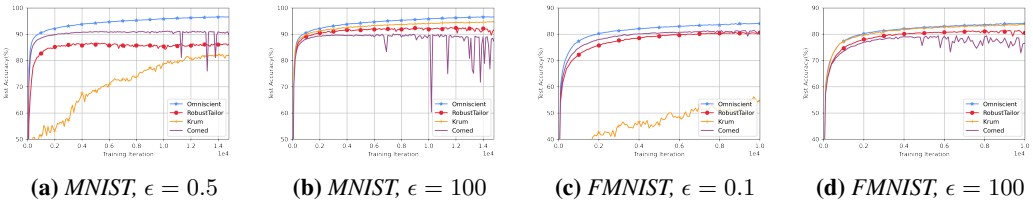

**(a)** *MNIST, $\epsilon = 0.5$*     **(b)** *MNIST, $\epsilon = 100$*     **(c)** *FMNIST, $\epsilon = 0.1$*     **(d)** *FMNIST, $\epsilon = 100$*

**Figure 1:** *Test accuracy on MNIST and FMNIST under iid setting. Tailored attacks ($\epsilon = 0.1/0.5, 100$) are applied.* RobustTailor *selects an aggregator from Krum and Comed based on the simulation at each iteration.*

**Mixed attacks.** We now consider additional and stronger attack strategies beyond vanilla $\epsilon$-reverse attacks. We assume the adversary has a set of attacks including $\epsilon = 0.5$ and $\epsilon = 100$ attacks. **StochasticAttack** shown in Fig. 2 picks an attack from its set *uniformly at random* at each iteration. **AttackTailor** in Fig. 3 optimizes an attack based on simulation at each iteration, whose detailed algorithm is in Appendix D. Compared to all previous attacks including StochasticAttack, AttackTailor is much stronger since it can pick a proper attack under perfect knowledge of honest updates. The poison of AttackTailor shown in Fig. 3 is almost as effective as the most targeted attack tailored against a single deterministic aggregator. Importantly, RobustTailor shows impressive robustness when facing such a strong adversary like AttackTailor.

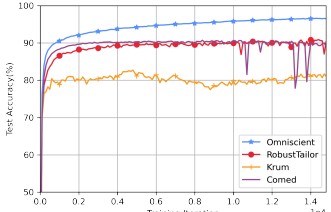
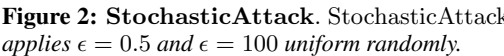
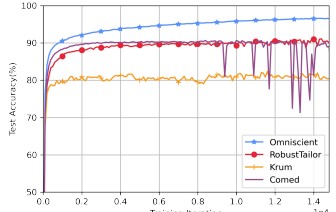

**Figure 2: StochasticAttack**. StochasticAttack *applies $\epsilon = 0.5$ and $\epsilon = 100$ uniform randomly.*

**Figure 3: AttackTailor**. AttackTailor *applies $\epsilon = 0.5$ and $\epsilon = 100$ based on the simulation.*

**Poisoned data mixed in the public dataset.** Byzantine clients may be able to donate poisoned data to the public dataset. We assume 16.7% of data in the public dataset is poisoned due to 16.7% of malicious clients. Two normal data poisoning methods we choose are label flipping (LF) (Muñoz-González et al., 2017) and random label (LR) (Zhang et al., 2021). Fig. 4 demonstrates that poisoned data mixed in has little impact on RobustTailor, which also proves that *a small gap* between the public dataset and true samples does not reduce the effectiveness of RobustTailor substantially.

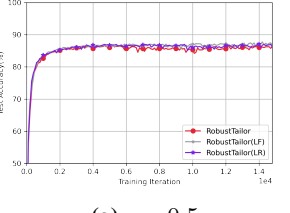 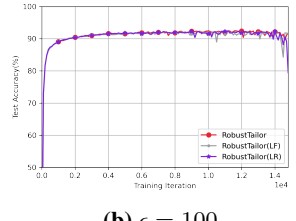 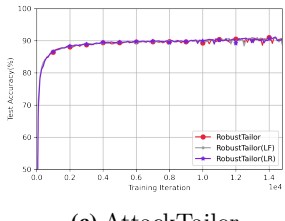

**(a)** $\epsilon = 0.5$        **(b)** $\epsilon = 100$        **(c)** AttackTailor

**Figure 4:** *Poisoned data mixed in the public dataset.*

**Unknown attacks for the server.** In our assumption, the server knows all attacks in the adversary's pool. What will happen if there is an attack out of the server's expectation? Fig. 5 gives the results. In particular, $\epsilon = 0.1$ in Fig. 5a and $\epsilon = 150$ in Fig. 5b are *the same type of attacks* as $\epsilon = 0.5/100$, and Mimic (Karimireddy et al., 2022) in Fig. 5c and Alittle (Baruch et al., 2019) in Fig. 5d are *the different type of attacks*. Note that we expand the set of RobustTailor with GM (Pillutla et al., 2022) and Bulyan (Guerraoui et al., 2018) as in Fig. 9 and decrease the learning rate to 0.005 for against Alittle and Mimic. It shows that RobustTailor can defend against not only the attacks similar to expectation but also those totally different. As a mixed framework, RobustTailor is hard to be ruined since the adversary hardly designs a tailored attack ruining several aggregation rules simultaneously.

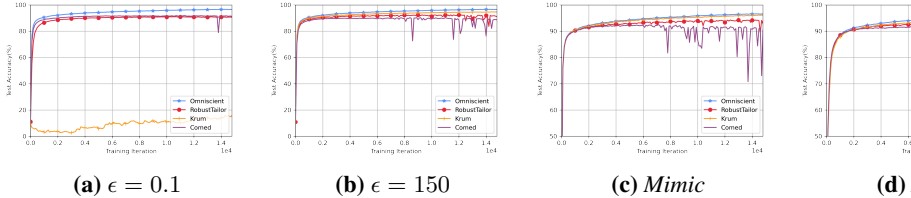

**(a)** $\epsilon = 0.1$     **(b)** $\epsilon = 150$     **(c)** *Mimic*     **(d)** *Alittle*

**Figure 5:** *Attacks out of the server's expectation.*

**Aggregators with auxiliary data.** Public dataset can be used not only for simulation but also to assist with aggregation. Fang et al. (2020) have proposed two server-side verification methods: error rate based rejection (ERR) and loss function based rejection (LFR), which reject potentially harmful gradients using error rates or loss values before aggregation. We provide experiments with the setup as in Fig. 3, and Krum/Comed assisted by ERR/LFR is totally ruined by AttackTailor with around **10%** accuracy while RobustTailor reaches **90.28%**. These results provide further evidence that RobustTailor delivers superior performance over existing techniques. By additional experiments, we observe that ERR/LFR helps aggregator achieve around 97% accuracy when facing $\epsilon = 0.5$ attack while it is totally ruined when against $\epsilon = 100$. In this more sensitive situation, AttackTailor is easily to break single aggregation rules but RobustTailor still performs well.

**Additional experiments.** To further validate the performance of RobustTailor, we set up additional experiments in Appendix G.2 including 1) three datasets; 2) non-iid settings; 3) more Byzantines; 4) more aggregation rules added in RobustTailor; 5) the impact of the proportion of public data; 6) subsampling by the server; 7) dynamic strategy of the adversary; 8) adversary with partial knowledge.

## 6   CONCLUSIONS AND FUTURE WORK

We formulate the robust distributed learning problem as a game between a server and an adversary. We propose RobustTailor, which achieves robustness by simulating the server's aggregation rules under different attacks optimized by an informed and powerful adversary. RobustTailor provides theoretical guarantees for the simulated game through a regret analysis. We empirically demonstrate the significant superiority of RobustTailor over baseline robust aggregation rules. Any Byzantine-resilient scheme with a given structure can be added to RobustTailor 's framework.

Although the increased computation complexity of RobustTailor is acceptable for the great robustness which is analyzed in Appendix H, it is also a future work to develop efficient and secure protocols to apply RobustTailor *e.g.,* using multi-party computation (Boneh et al., 2019).

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

**Notation.** We use $\mathbb{E}[\cdot]$, $\|\cdot\|$, $\|\cdot\|_0$, and $\|\cdot\|_*$ to denote the expectation operator, Euclidean norm, number of nonzero elements of a vector, and dual norm, respectively. We use $|\cdot|$ to denote the length of a binary string, the length of a vector, and cardinality of a set. We use lower-case bold letters to denote vectors. Sets are typeset in a calligraphic font. The base-2 logarithm is denoted by $\log$, and the set of binary strings is denoted by $\{0, 1\}^*$. We use $[n]$ to denote $\{1, \cdots, n\}$ for an integer $n$. We use $\Delta_M$ to denote the probability simplex in $\mathbb{R}^M$.

## A    COMPLETE RELATED WORK

**Federated learning (FL).** FL (McMahan et al., 2017; Konečnỳ et al., 2016) keeps training data decentralized in multiple clients which collaboratively train a model under the orchestration of a server (Kairouz et al., 2021). For the server, such clients are often more unpredictable and especially more vulnerable to the attacks. Secure aggregation protocols (Bonawitz et al., 2017; So et al., 2020) ensure that the server computes aggregated updates without revealing the original data. In this paper, we focus on training-time attacks and corresponding aggregation rules.

**Training-time attacks.** Standard attacks can be broadly classified into training-time attacks (poisoning attacks) (Biggio et al., 2012; Li et al., 2016; Jagielski et al., 2018; Bhagoji et al., 2019; Huang et al., 2011; Mei & Zhu, 2015; Alfeld et al., 2016; Koh & Liang, 2017; Mahloujifar et al., 2019; Gu et al., 2019; Xie et al., 2019; Wang et al., 2020b; Yang & Li, 2021; Karimireddy et al., 2021; Data & Diggavi, 2021; Carlini & Terzis, 2021; Allen-Zhu et al., 2021) and inference-time attacks (evasion attacks) (Goodfellow et al., 2014; Carlini & Wagner, 2017). Because the server in FL trains the model across various unreliable clients with private datasets, FL usually suffers from training-time attacks (Biggio et al., 2012; Bhagoji et al., 2019; Sun et al., 2019; Bagdasaryan et al., 2020). A strong adversary can potentially participate in every training round, and meanwhile it can adapt its attacks to an updated model. One class of training-time attacks concerned in this work is *model update poisoning*. In model poisoning attack, an adversary can control some clients and can directly manipulate their outputs trying to bias the global model towards the opposite direction (Kairouz et al., 2021). If Byzantine clients have access to the updates of honest clients, they can tailor their attacks and make them difficult to detect (Fang et al., 2020; Xie et al., 2020; Lamport et al., 1982; Blanchard et al., 2017; Goodfellow et al., 2014; Bagdasaryan et al., 2020).

**Robust aggregation and Byzantine resilience.** To improve robustness under general Byzantine clients, a number of robust aggregation schemes have been proposed, which are mainly inspired by robust statistics such as median-based aggregators (Yin et al., 2018; Chen et al., 2017), Krum (Blanchard et al., 2017), trimmed mean (Yin et al., 2018). Krum (Blanchard et al., 2017) and coordinate-wise median (Comed) (Yin et al., 2018; Chen et al., 2017) are two main aggregation rules used in this paper. Krum is a squared-distance-based aggregation rule and it aggregates the gradients that minimize the sum of squared distances to its $n - f - 2$ closest vectors where $n$ denotes the total number of clients and $f$ is the number of adversarial ones. Comed is a median-based aggregator and it selects the gradient closest to the median of each dimension.

Except of statistical aggregation rules, there are still many related works like server-side verification, client-side self-clipping etc. From the perspective of the server, Fang et al. (2020); Xie et al. (2020); Cao & Lai (2019); Cao et al. (2020) propose some server-side verification methods using auxiliary data. Specifically, Fang et al. (2020) assume the server has a small validation dataset and uses error rates to reject harmful gradients. In (Xie et al., 2020; Cao & Lai, 2019), the server asks a small clean dataset from clients and filters out unreliable gradients. Cao et al. (2020) utilize the ReLU-clipped cosine-similarity between local gradients and the standard one calculated by a small clean dataset as the weight for aggregation. Moreover, Karimireddy et al. (2021) and Alistarh et al. (2018) propose history-aided aggregators, and an expandable framework proposed by Ramezani-Kebrya et al. (2022) utilizes randomization to improve robustness. None of them selects a proper aggregation rule proactively during training as our framework $\mathrm{RobustTailor}$. We note that all aggregation rules shown here can be added to the pool of $\mathrm{RobustTailor}$ because a public dataset is available in our assumption and any aggregation rule can use it. In addition, client-side clipping methods are proposed by Sun et al. (2021) and Sun et al. (2019), and client-side momentum SGD is considered by Karimireddy et al. (2021) and El Mhamdi et al. (2021). However, the ability of clients is not the focus of our paper and we will consider it in future work.

Past work has shown that these aggregators can defend successfully under specific conditions (Blanchard et al., 2017; Chen et al., 2017; Su & Vaidya, 2016). However, Fang et al. (2020) and Xie et al. (2020) argue that Byzantine-resilient aggregators can fail when an informed adversary tailors a careful attack. Therefore, developing a robust and efficient algorithm under such strong tailored attacks is essential to improve security of FL, which is the goal of this paper.

**Heterogeneous data.** In the real world applications, many issues such as heterogeneous data become significant (Kairouz et al., 2021; Karimireddy et al., 2020). Karimireddy et al. (2022) find that robust learning algorithms of FL may fail under iid setting. Several algorithms are proposed to tackle non-iid data (Yoshida et al., 2020; Zhao et al., 2018; Karimireddy et al., 2022; Wang et al., 2020a; Data & Diggavi, 2021; Zhu et al., 2021). Besides, data heterogeneity easily leads to backdoor attacks, which can be viewed as a kind of training-time attacks (Xie et al., 2019; Bagdasaryan et al., 2019; Zawad et al., 2021). Therefore, establishing robustness under non-iid setting is also an important indicator for an aggregator.

**Game theory in FL.** *Online Convex Optimization* (OCO) framework (Zinkevich, 2003) is widely influential in the learning community (Hazan et al., 2016; Shalev-Shwartz et al., 2012), and *Bandit Convex Optimization* (BCO) as an extension of OCO was proposed by Awerbuch & Kleinberg (2008) for decision making with limited feedback. Bandit paradigms paralleling FL framework are proposed by Shi & Shen (2021) and its extension under Byzantine attacks is proposed by Demirel et al. (2022). However, they account for uncertainties from both arm and client sampling rather than robust aggregation in this paper. In this paper, we frame robust distributed learning problem as a game and consider the *bandit feedback model*.

# B ROBUSTNESS OF RobustTailor

In this section, we define a general robustness definition of an aggregation rule against an attack. Note that our definition covers a broad range of settings with *general pure and mixed aggregation* along with *general pure and mixed attack* strategies. Our robustness notion leads to almost sure convergence guarantees to a local minimum of $F$ in FL, which is equivalent to being immune to training-time attacks.

**Definition 4** (Robustness of an aggregator to an attack program). *Let $\mathbf{x} \in \mathbb{R}^d$ denote a machine learning model. Let $\mathbf{g}_i(\mathbf{x}) = \nabla F_i(\mathbf{x}) \in \mathbb{R}^d$ be independent honest updates for $i \in [n]$. Let $G(\mathbf{x})$ denote a function that draws an honest client $i$ uniformly at random followed by outputting an unbiased stochastic gradient of $\nabla F_i(\mathbf{x})$ over that client such that $\mathbb{E}[G(\mathbf{x})] = \nabla F(\mathbf{x})$ where $\mathbb{E}$ is over both random client and samples. Let $\mathrm{AG}$ denote an arbitrary aggregation rule, which can be a mixed aggregation strategy selecting an aggregator from $\mathcal{A} = \{\mathrm{AG}_1, \ldots, \mathrm{AG}_M\}$ based on simulation. The output of $\mathrm{AG}$ is given by $\breve{g}(\mathbf{x}) = \mathrm{AG}(\{\mathbf{g}'\}_{i=1}^n)$. Note that $\{\mathbf{g}'\}_{i=1}^n$ includes both honest and compromised updates. The compromised updates are the output of an attack program $\mathrm{AT}(\{\mathbf{g}_i\}_{i=1}^{n-f}, \mathcal{A})$. Note that $\mathrm{AT}$ can be a pure or mixed attack strategy.*

*The mixed aggregation rule $\mathrm{AG}$ is Byzantine-resilient to $\mathrm{AT}$ if $\breve{g}(\mathbf{x})$ satisfies $\mathbb{E}[\breve{g}(\mathbf{x})]^\top \nabla F(\mathbf{x}) > 0$ and $\mathbb{E}[||\breve{g}(\mathbf{x})||^r] \leq K_r \mathbb{E}[||G(\mathbf{x})||^r]$ for $r = 2, 3, 4$ and some constant $K_r$.*

Suppose $\{\eta_t\}_{i=1}^\infty$ in Algorithm 3 satisfies $\sum_t \eta_t = \infty$ and $\sum_t \eta_t^2 < \infty$. For a *nonconvex loss function*, which is three times differentiable with continuous derivatives, bounded from below, and satisfies global confinement assumption in (Bottou, 1998, Section 5.1), *general pure and mixed aggregation and attack strategies* satisfying Definition 4, and *general non-iid data distribution across clients*, we can establish almost sure convergence ($\nabla F(\mathbf{x}_t) \to 0$ a.s.) of the output of AG in Algorithm 3 along the lines of (Bottou, 1998; Fisk, 1965; Métivier, 1982).

Note that to achieve $\mathbb{E}[\breve{g}(\mathbf{x})]^\top \nabla F(\mathbf{x}) > 0$ shown above, it requires both the distance between $\nabla F(\mathbf{x})$ and the estimate of the honest update $\widetilde{\mathbf{g}}$ and the distance between $\widetilde{\mathbf{g}}$ and the expected output of Algorithm 2, i.e., $\mathbb{E}[\breve{g}(\mathbf{x})]$, are small. Let $\theta_1$ denote the angle between $\nabla F(\mathbf{x})$ and $\widetilde{\mathbf{g}}$, and let $\theta_2$ denote the angle between $\widetilde{\mathbf{g}}$ and $\mathbb{E}[\breve{g}(\mathbf{x})]$, given by $\arg\cos\left(\frac{\widetilde{\mathbf{g}}^\top \nabla F(\mathbf{x})}{\|\widetilde{\mathbf{g}}\| \cdot \|\nabla F(\mathbf{x})\|}\right)$ and $\arg\cos\left(\frac{\widetilde{\mathbf{g}}^\top \mathbb{E}[\breve{g}(\mathbf{x})]}{\|\widetilde{\mathbf{g}}\| \cdot \|\mathbb{E}[\breve{g}(\mathbf{x})]\|}\right)$, respectively. If $\theta_1 + \theta_2 < \pi/2$, then we have $\mathbb{E}[\breve{g}(\mathbf{x})]^\top \nabla F(\mathbf{x}) > 0$. Following the arguments in Appendix B, almost sure convergence of Algorithm 3 is guaranteed as long as $\theta_1 + \theta_2 < \pi/2$. This condition can be satisfied assuming 1) the public data donated by clients is representative of the underlying data distribution of honest clients, which controls $\theta_1$, and 2) the number of Byzantine

clients is sufficiently small, which controls $\theta_2$. We defer derivation of the explicit necessary condition for almost sure convergence to future work.

## C    DETAILS OF EXP3

The bandit feedback model considers the following iterate game.

**Definition 5** (Bandit setting). *The player is given a decision set $[N]$. At each iteration $k = 1, \ldots, K$:*

1. *the player picks $i_k \in [N]$.*

2. *the adversary picks a loss vector $\ell^k$.*

3. *the player observes and suffers the loss at index $i_k$, i.e. $\ell^k(i_k)$.*

Exp3, as shown abstractly in Algorithm 4, enjoys a so called no-regret property in this setting. We employ Exp3 from both the perspective of a simulated server and simulated attacker to find a robust aggregation rule in Algorithm 2. In Appendix E we show how to convert the no-regret properties into a convergence guarantee.

---

**Algorithm 4** Exp3

---

**Input:** Updating rate $\lambda$ and $\tilde{\lambda}$, iteration rounds $K$, $N$

1 Initialize weight vector $w^0(i) = 1$ for $i = 1, \ldots, N$.

2 **for** $k = 1$ **to** $K$ **do**

3      Set $W^k = \sum_{i=1}^N w^k(i)$, and set for $i = 1, \ldots, N$

$$\mathbf{p}(i) = (1 - \lambda)\frac{w^k(i)}{W^k} + \lambda\frac{1}{N}$$

4      Draw $i_k$ randomly accordingly to the probabilities $\mathbf{p}$.

5      Receive loss $\ell^k$.

6      Set for $i = 1, \ldots, N$

$$\hat{\ell}^k(i) = \begin{cases} \ell^k/\mathbf{p}(i), & \text{if } i = i_k; \\ 0, & \text{otherwise.} \end{cases}$$

$$w^{k+1}(i) = w^k(i) \exp(-\tilde{\lambda}\hat{\ell}^k(i)/N).$$

---

## D    SIMULATION OF ADVERSARY

In this section, we show the simulation of the adversary. We term *adversarial simulation* as AttackTailor, which outputs an appropriate AT at each iteration. The specific steps from the perspective of the adversary is shown in Algorithm 5. After observing $n - f$ honest gradients, the server performs $K$-round simulation and obtains a final probability distribution $\mathbf{q}$. By sampling from $\mathbf{q}$, the server selects an attack. Then, $f$ Byzantine clients create and send the compromised gradients to the server. The steps for simulating the attack procedure are summarized in Algorithm 6.

Importantly, the main difference between the adversary's simulation compared with that of server is that the adversary does simulation based on realistic honest gradients while the server has access only to *noisy estimates* of true gradients. Hence, unlike typical games and simulation setups, the adversary has an additional advantage over the server, which is due to *information asymmetry*.

---

**Algorithm 5** AttackTailor

---

**Input:** Updating rates $\lambda_1$, $\lambda_2$, $\tilde{\lambda}_1$ and $\tilde{\lambda}_2$, simulation rounds $K$, gradients of honest clients $\{\mathbf{g}_i\}_{i=1}^{n-f}$, $\mathcal{A}$ and $\mathcal{F}$

1   Initialize weight vector $w_1^0(i) = 1$ for $i \in [M]$ and $w_2^0(j) = 1$ for $j \in [S]$.

    **for** $k = 1$ **to** $K$ **do**

2       Set $\mathbf{p}^k(\widetilde{\mathrm{AG}_i}) = (1 - \lambda_1)\frac{w^k(i)}{\sum_{i=1}^M w^k(i)} + \lambda_1 \frac{1}{M}$ for $i = 1, \ldots, M$.

3       Set $\mathbf{q}^k(\widetilde{\mathrm{AT}_j}) = (1 - \lambda_2)\frac{v^k(j)}{\sum_{j=1}^S v^k(j)} + \lambda_2 \frac{1}{S}$ for $j = 1, \ldots, S$.

4       Sample $\mathrm{AG}^k \sim \mathbf{p}^k$ and $\mathrm{AT}^k \sim \mathbf{q}^k$ respectively.

5       Estimate the loss $\ell^k = \tilde{\ell}(\mathrm{AG}^k, \mathrm{AT}^k, \{\mathbf{g}_i\}_{i=1}^{n-f})$.

6       Set for $i = 1, \ldots, M$

$$\hat{\ell}_1^k(i) = \frac{\mathbb{I}\{\widetilde{\mathrm{AG}_i} = \mathrm{AG}^k\}}{\mathbf{p}^k(\widetilde{\mathrm{AG}_i})}\ell^k, \quad w_1^{k+1}(i) = w_1^k(i)\exp(-\tilde{\lambda}_1 \hat{\ell}_1^k(i)/M).$$

7       Set for $j = 1, \ldots, S$

$$\hat{\ell}_2^k(j) = \frac{\mathbb{I}\{\widetilde{\mathrm{AT}_j} = \mathrm{AT}^k\}}{\mathbf{q}^k(\widetilde{\mathrm{AT}_j})}\ell^k, \quad w_2^{k+1}(j) = w_2^k(j)\exp(\tilde{\lambda}_2 \hat{\ell}_2^k(j)/S).$$

8  Set for $i = 1, \ldots, M$

$$\mathbf{q}_j = \frac{\sum_{k=1}^K \mathbf{q}^k(\mathrm{AT}_j)}{K}.$$

9  Sample $\mathrm{AT} \sim \mathbf{q}$.

  **Output:** AT.

---

**Algorithm 6** Adversary's attack

---

**Input:** Learning rate $\eta_t$, $n$ workers, $f$ compromised workers, iteration rounds $T$, $\mathcal{A}$ and $\mathcal{F}$

1   **for** $t = 1$ **to** $T$ **do**

2       Observe all gradients of honest workers $\{\mathbf{g}_i\}_{i=1}^{n-f}$.

3       Call Algorithm to attack $\mathrm{AT}^t = \mathrm{AttackTailor}(\{\mathbf{g}_i\}_{i=1}^{n-f}, \mathcal{A}, \mathcal{F})$.

4       Produce $f$ gradients for compromised clients. Set for $j \in [f]$

$$\mathbf{b}_j = \mathrm{AT}^t(\{\mathbf{g}_i\}_{i=1}^{n-f}, \mathcal{A}).$$

5       Send compromised gradients $\{\mathbf{b}_j\}_{j=1}^f$ to the server.

---

## E   PROOF OF LEMMA 1

*Proof.* Defined $\bar{i} \sim \bar{\mathbf{p}}$ to be uniformly sampled from $\{i_k\}_{k=1}^K$, and $\bar{j} \sim \bar{\mathbf{q}}$ to be uniformly sampled from $\{j_k\}_{k=1}^K$. Using the no-regret property from (8),

$$\mathbb{E}[L(\bar{i}, j)] = \mathbb{E}\left[\frac{1}{K}\sum_{k=1}^K L(i_k, j)\right] \leq \mathbb{E}\left[\frac{1}{K}\sum_{k=1}^K L(i_k, j_k)\right] + \frac{1}{K}\mathcal{R}_K^{\mathrm{j}}$$

$$\mathbb{E}[L(i, \bar{j})] = \mathbb{E}\left[\frac{1}{K}\sum_{k=1}^K L(i, j_k)\right] \geq \mathbb{E}\left[\frac{1}{K}\sum_{k=1}^K L(i_k, j_k)\right] - \frac{1}{K}\mathcal{R}_K^{\mathrm{i}},$$

(12)

for any $i \in [M]$ and $j \in [S]$, where the expectation is taken over $\bar{i}$ and $\bar{j}$ and the randomness of the algorithms. Subtracting the two equations,

$$\mathbb{E}[L(\bar{i}, j)] - \mathbb{E}[L(i, \bar{j})] \leq \frac{1}{K}(\mathcal{R}_K^{\mathrm{j}} + \mathcal{R}_K^{\mathrm{i}}) =: \varepsilon. \tag{13}$$

Observe that by first evoking the inequality with $i \sim \bar{\mathbf{p}}$ and secondly with $j \sim \bar{\mathbf{q}}$, we see that $(\bar{\mathbf{p}}, \bar{\mathbf{q}})$ is an $\varepsilon$-approximate Nash equilibrium, i.e.,

$$\mathbb{E}[L(\bar{i}, j)] - \varepsilon \leq \mathbb{E}[L(\bar{i}, \bar{j})] \leq \mathbb{E}[L(i, \bar{j})] + \varepsilon. \tag{14}$$

We are interested in the $i$-players performance $\mathbb{E}[L(\bar{i}, j)]$ which we can relate to the mixed strategy Nash equilibrium defined as $\mathbb{E}[L(i^\star, j)] \leq \mathbb{E}[L(i^\star, j^\star)] \leq \mathbb{E}[L(i, j^\star)]$ where $i^\star \sim \mathbf{p}^\star$ and $j^\star \sim \mathbf{q}^\star$. By picking $i \sim \mathbf{p}^\star$ in (13) we get,

$$
\begin{aligned}
\mathbb{E}[L(\bar{i}, j)] &\leq \mathbb{E}[L(i, \bar{j})] + \frac{1}{K}(\mathcal{R}_K^{\mathrm{j}} + \mathcal{R}_K^{\mathrm{i}}) \\
&= \mathbb{E}[L(i^\star, \bar{j})] + \frac{1}{K}(\mathcal{R}_K^{\mathrm{j}} + \mathcal{R}_K^{\mathrm{i}}) \\
&\leq \mathbb{E}[L(i^\star, j^\star)] + \frac{1}{K}(\mathcal{R}_K^{\mathrm{j}} + \mathcal{R}_K^{\mathrm{i}}),
\end{aligned}
\tag{15}
$$

where the last inequalities follows by the definition of a Nash equilibrium above. The claim follows by writing the expectation on the RHS in terms of $\mathbf{p}^\star$ and $\mathbf{q}^\star$. $\qquad\square$

## F   PROOF OF LEMMA 3

*Proof.* Let both player $i$ and player $j$ in Lemma 1 employ the no-regret algorithm Exp3 such that Lemma 2 applies and consequently $\mathcal{R}_K^{\mathrm{i}}$ and $\mathcal{R}_K^{\mathrm{j}}$ in (8) reduce to

$$
\begin{aligned}
\mathcal{R}_K^{\mathrm{i}} &= 2\sqrt{\mathrm{KM}\log M} \\
\mathcal{R}_K^{\mathrm{j}} &= 2\sqrt{\mathrm{KS}\log S}.
\end{aligned}
\tag{16}
$$

Substituting (16) into Lemma 1, we have

$$
\mathbb{E}_{\bar{i}}\left[L(\bar{i}, j)\right] \leq \mathbb{E}_{i^\star \sim \mathbf{p}^\star, j^\star \sim \mathbf{q}^\star}\left[L(i^\star, j^\star)\right] + 2\frac{\sqrt{M\log M} + \sqrt{S\log S}}{\sqrt{K}}.
\tag{17}
$$

Notice that Algorithm 2 is an instance of two simultaneously played Exp3 algorithms where $i = \mathrm{AG}$, $j = \mathrm{AT}$ and $L(i, j) = \widetilde{\ell}\left(\mathrm{AG}, \mathrm{AT}, \{\mathbf{g}_i'\}_{i=1}^n\right)$. It follows from (17) that

$$
\begin{aligned}
\mathbb{E}_{\mathrm{AG}, \mathcal{G}}\left[\widetilde{\ell}\left(\mathrm{AG}, \mathrm{AT}, \{\mathbf{g}_i'\}_{i=1}^n\right)\right] &\leq \mathbb{E}_{\mathrm{AG}^\star \sim \mathbf{p}^\star, \mathrm{AT}^\star \sim \mathbf{q}^\star, \mathcal{G}}\left[\widetilde{\ell}(\mathrm{AG}^\star, \mathrm{AT}^\star, \{\mathbf{g}_i'\}_{i=1}^n)\right] \\
&\quad + 2\frac{\sqrt{M\log M} + \sqrt{S\log S}}{\sqrt{K}}
\end{aligned}
\tag{18}
$$

where $\mathrm{AG}$ is the average iterate as defined in Algorithm 2. We can concisely write the Nash equilibrium on the R.H.S. of (18) in terms of the payoff matrix $\mathcal{L}$ from Sim-MinMax defined componentwise as $\mathcal{L}(\mathrm{AG}, \mathrm{AT}) = \widetilde{\ell}(\mathrm{AG}, \mathrm{AT}, \{\mathbf{g}_i'\}_{i=1}^n)$. This completes the proof. $\qquad\square$

## G   EXPERIMENTAL DETAILS AND ADDITIONAL EXPERIMENTS

In this section, we provide the training hype-parameters and show a series of additional experiments.

### G.1   DETAILS OF IMPLEMENTATION

Both MNIST (Lecun et al., 1998) and FMNIST (Xiao et al., 2017) datasets contain 60000 training samples and 10000 test samples. Each sample is a 28 by 28 pixel grayscale image. The details of training hype-parameters are shown in Table 1. The network architecture is a fully connected neural network with two fully connected layers (Leroux et al., 2016). The number of neurons is 100 and 10 for the first and second layer, respectively. All experiments have been run on a cluster with Xeon-Gold processors and V100 GPUs.

### G.2   ADDITIONAL EXPERIMENTS

To further validate the performance of $\mathrm{Robust Tailor}$, we set up additional experiments:

- Training on 3 datasets.
- Non-iid settings with 3 different heterogeneous degree.

**Table 1:** Training Hyper-parameters for Fashion-MNIST and MNIST

| Hyper-parameter | MNIST | FMNIST | CIFAR10 |
|---|---|---|---|
| Learning Rate | 0.01 | 0.003 | 0.002 |
| Batch Size | 50 | 50 | 80 |
| Total Iterations | 15K | 10K | 10K |
| Simulation Rounds | 10 | 10 | 10 |
| $\lambda_1, \lambda_2$ | 0.3 | 0.3 | 0.3 |
| $\tilde{\lambda}_1, \tilde{\lambda}_2$ | 0.3 | 0.3 | 0.3 |

- More Byzantines.
- More aggregators (Geomed, Trimmedmean, Bulyan) added against single attack.
- The impact of the proportion of public data.
- Subsampling by the server.
- Dynamic strategy of the adversary.
- Adversary with partial knowledge.

**Different datasets.** We train a CNN model on MNIST (Lecun et al., 1998), Fashion-MNIST (FMNIST) (Xiao et al., 2017) and CIFAR10 (Krizhevsky et al., 2009) under iid setting. We summarize the training results against 3 attacks here and they are consensus with the results shown in Section 5.

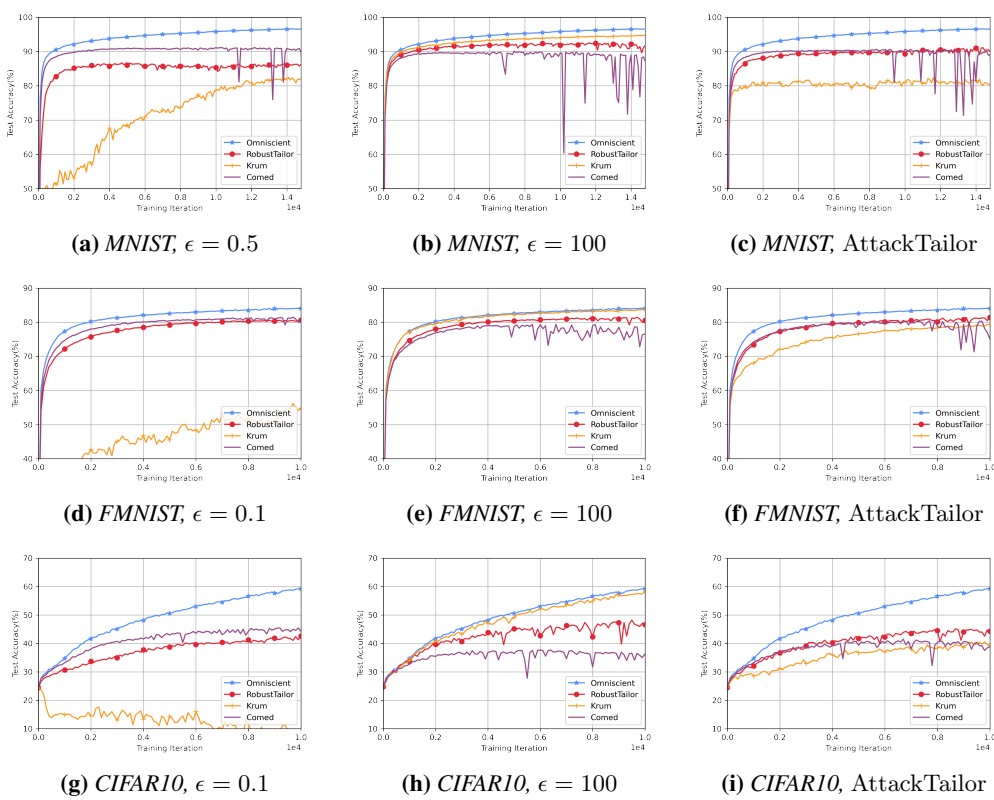

**(a)** *MNIST,* $\epsilon = 0.5$     **(b)** *MNIST,* $\epsilon = 100$     **(c)** *MNIST,* AttackTailor

**(d)** *FMNIST,* $\epsilon = 0.1$     **(e)** *FMNIST,* $\epsilon = 100$     **(f)** *FMNIST,* AttackTailor

**(g)** *CIFAR10,* $\epsilon = 0.1$     **(h)** *CIFAR10,* $\epsilon = 100$     **(i)** *CIFAR10,* AttackTailor

**Figure 6:** *iid setting on three datasets.* RobustTailor *includes Krum and Comed.* AttackTailor *includes* $\epsilon = 0.1/0.5$ *and* $\epsilon = 100$.

**Non-iid settings.** We also extend our consideration to more realistic settings with non-iid data distribution across clients. We use the *heterogeneous degree* $\mu \in [0, 1]$ to represent the level of disparity among clients' local data. To be specific, we construct a setting, in which $100\mu$ % of local data for each client is drawn in a non-identical but independent manner from a particular class

corresponding to the client index and $100(1 - \mu)$ % of the local data is drawn iid from all classes. A small $\mu$ represents low disparity while a large $\mu$ means significant disparity among clients. Fig. 7 shows three non-iid settings including $\mu = 0.1, 0.5, 0.9$. We surprisingly observe that RobustTailor shows a satisfactory level of robustness even under heterogeneous data settings.

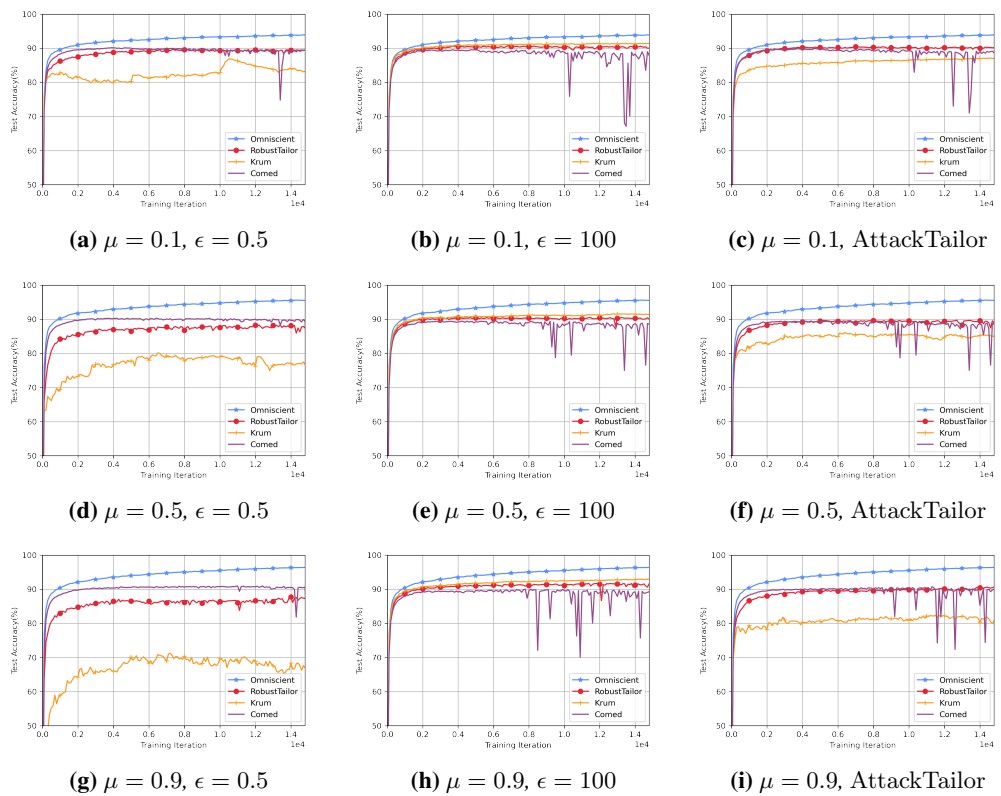

**Figure 7:** *Non-iid setting. Larger $\mu$ means higher heterogeneous degree.*

**More Byzantines.** Fig. 8 shows the results when there are 4 Byzantines in 12 total clients under three different attacks. Except the number of compromised clients, which is four instead of two, the setting is same as that in Fig. 1. We observe that both Krum and Comed are sensitive to the number of Byzantine clients while RobustTailor is much more stable. Specifically, Krum has lower accuracy closing to zero, and Comed shows more obvious fluctuations.

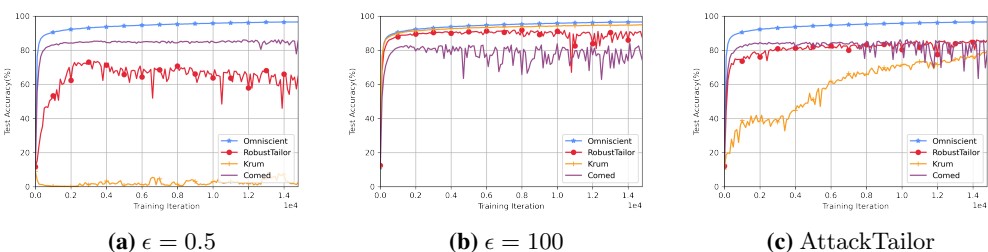

**Figure 8:** *4 Byzantines. There are $n = 12$ total workers including $f = 4$ Byzantine workers.*

**More aggregators against single attacks.** To show the intuitive results of RobustTailor, we just construct the server's pool with two aggregators including Krum and Comed in the main text. However, RobustTailor could outperform both Krum and Comed simultaneously when additional aggregators are put into the server's pool. Trimmedmean (TM) (Yin et al., 2018), Geomed (GM) (Pillutla et al., 2022), and Bulyan (Guerraoui et al., 2018) are also statistic-based Byzantine-resilient aggregators

and all of them can be added in RobustTailor. When Bulyan is included, each Bulyan aggregator uses a different aggregator from Krum, Comed, TM, and GM for either the selection phase or in the aggregation phase. For each class, we generate 16 aggregators, each with a randomly generated $\ell_p$ norm from one to 16. RobustTailor selects one aggregator from the entire pool of 64 aggregators based on the simulation at each iteration. Moreover, centered clipping (CC) (Karimireddy et al., 2021) as a history-aided aggregator was proposed recently and it can also be added in RobustTailor framework. Fig. 9 shows results when more aggregators added into RobustTailor framework and they perform better.

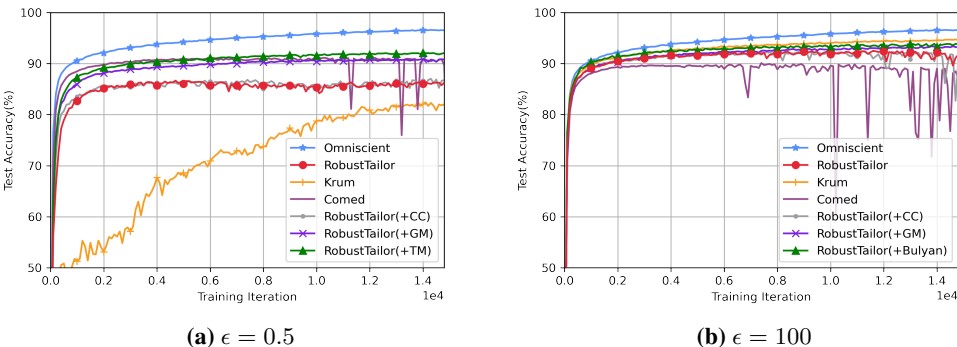

**(a)** $\epsilon = 0.5$        **(b)** $\epsilon = 100$

**Figure 9:** *More aggregators added into to* RobustTailor *structure against single attacks.*

**Proportion of public data.** Because RobustTailor ask for clients to donate a small amount of data as the public dataset, we want to minimise data leakage as much as possible while maintaining effectiveness. Therefore, detecting the impact of the proportion of public data is necessary. In our experiments, we assume every client donates 5% of data to the server. Fig. 10 shows the performance of RobustTailor with different proportion of public data under 3 attacks. Note that except for the proportion of public data, other settings of 3 figures in Fig. 10 are the same as Fig. 1a, Fig. 1b, and Fig. 3 respectively. It is obvious that the amount of public data has little impact on RobustTailor and even very small proportion of data donated by clients (*e.g.,* 0.1%) can help RobustTailor achieve a great performance. This also strongly proves Remark 1 that the public dataset just need to represent clients' data distribution.

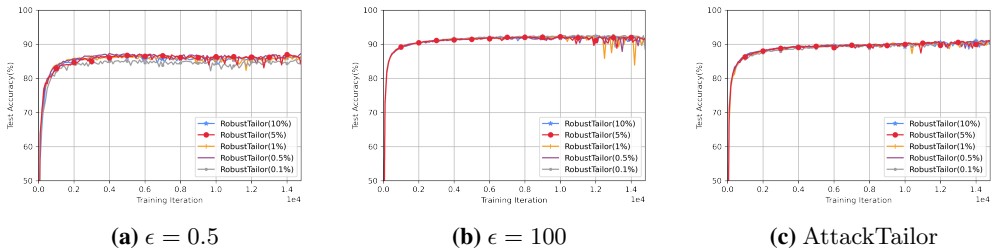

**(a)** $\epsilon = 0.5$       **(b)** $\epsilon = 100$       **(c)** AttackTailor

**Figure 10:** *The impact of the proportion of public data.*

**Subsampling by the server.** Subsampling, a common technique in large-scale FL, can also increase the complexity of attacks. The server picks a subset of clients randomly and asks for updates at each iteration. The adversary can know which clients are selected and leverage the selected honest updates to design the attack for the compromised clients which are also chosen. Due to subsampling usually used in FL with large amount of clients, we run additional experiments with 120 clients, in which 20 clients are Byzantines. The server chooses 10% of clients randomly for aggregation at each iteration, and it assumes that 2 of 12 clients are compromised every iteration. Note that other settings of experiments are same as Fig. 1a, Fig. 1b, and Fig. 3 in the paper except for learning rate. We decrease the learning rate from 0.01 to 0.001 because all aggregation methods are too unstable under the original setting. The results are shown in Fig. 11, which is consistent with the results without subsampling.

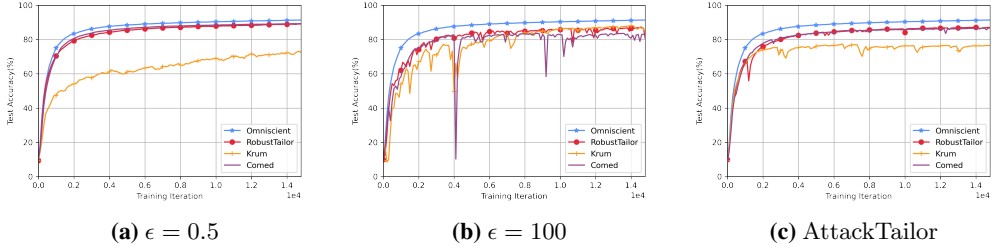

**Figure 11:** *Subsampling by the adversary.*

**Dynamic strategy of the adversary.** The adversary can also use a dynamic strategy that changes the number of malicious updates dynamically. The adversary picks 1-3 clients randomly to control at each iteration while the server still considers 2 Byzantines in 12 clients. Other settings are the same as Section 5. Fig. 12 shows the results and Table 2 compares them with the results without dynamic attack strategy in Section 5. Although some aggregation rules are impacted lightly by such dynamic attack strategy, RobustTailor has a good performance which is consistent with the original results.

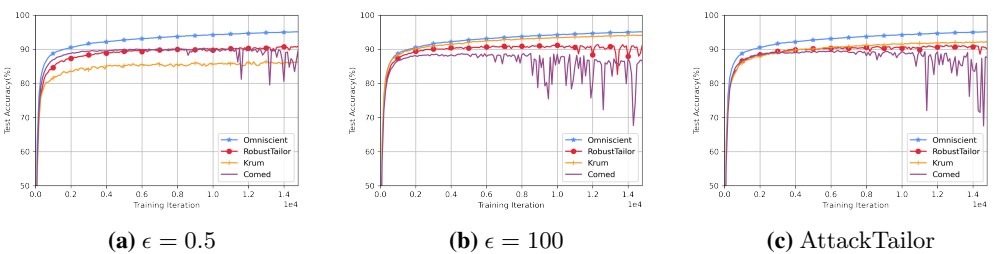

**Figure 12:** *Dynamic attack strategy of the adversary.*

**Table 2:** Comparison between the adversary with and without dynamic strategy.

| Attack | Aggregator | With dynamic attack | Without dynamic attack |
|--------|-----------|---------------------|------------------------|
| | RobustTailor | 87.49% | 90.54% |
| $\epsilon = 0.5$ | Krum | 84.37% | 82.13% |
| | Comed | 72.80% - 90.57% | 72.07% - 90.74% |
| | RobustTailor | 85.26% - 91.62% | 91.72% |
| $\epsilon = 100$ | Krum | 93.88% | 94.74% |
| | Comed | 71.80% - 88.15% | 71.72% - 89.01% |
| | RobustTailor | 90.17% | 90.54% |
| AttackTailor | Krum | 75.75% | 81.68% |
| | Comed | 77.37% - 88.05% | 74.69% - 89.69% |

**Adversary with partial knowledge.** What we want to demonstrate in the main text is that RobustTailor can keep a great and stable performance even when facing a very strong adversary who has full knowledge of all honest clients. However, it is very hard for the adversary to have full knowledge of the model updates of all honest clients in reality. Although Fang et al. (2020) show that the partial knowledge attacks are weaker than the full knowledge attack's, it is still significant to consider more realistic attacks. For the partial knowledge setting, assume the adversary only knows the updates of two honest clients and design compromised gradients based on them. Note that other settings are the same as Section 5. We show the empirical results under both iid and non-iid (the heterogenous degree $\mu = 0.9$) settings in Fig. 13 and compares them against the adversary with full knowledge in Table 3. Most aggregation rules can perform at least the same as the scenario of the adversary with full knowledge.

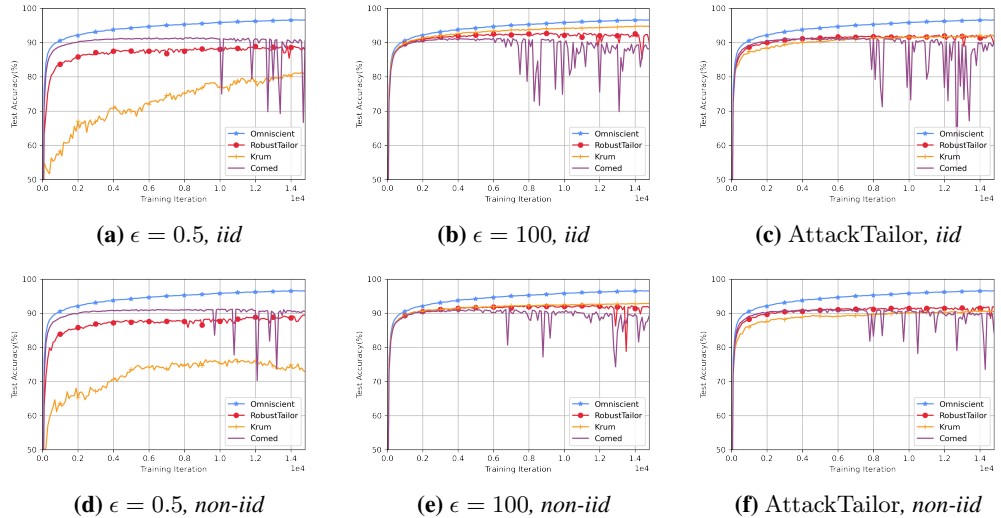

**Figure 13:** *Adversary with partial knowledge.*

**Table 3:** Comparison between the adversary partial knowledge with and with full knowledge.

**(a)** $\epsilon = 0.5$

| Attack | Aggregator | Partial | Full |
|--------|-----------|---------|------|
| iid | RobustTailor | 89.27% | 90.54% |
| | Krum | 79.58% | 82.13% |
| | Comed | 66.69% - 90.65% | 72.07% - 90.74% |
| non-iid | RobustTailor | 88.93% | 87.54% |
| | Krum | 73.50% | 66.96% |
| | Comed | 80.85% - 90.54% | 81.84% - 90.48% |

**(b)** $\epsilon = 100$

| Attack | Aggregator | Partial | Full |
|--------|-----------|---------|------|
| iid | RobustTailor | 85.18% - 92.31% | 91.72% |
| | Krum | 94.74% | 94.74% |
| | Comed | 69.83% - 89.32% | 71.72% - 89.01% |
| non-iid | RobustTailor | 86.27% - 92.03% | 90.62% |
| | Krum | 92.85% | 92.85% |
| | Comed | 73.71% - 89.75% | 75.73% - 89.38% |

**(c)** AttackTailor

| Attack | Aggregator | Partial | Full |
|--------|-----------|---------|------|
| iid | RobustTailor | 91.67% | 90.54% |
| | Krum | 91.77% | 81.68% |
| | Comed | 67.13% - 90.06% | 74.69% - 89.69% |
| non-iid | RobustTailor | 91.51% | 90.26% |
| | Krum | 90.60% | 80.97% |
| | Comed | 73.53% - 90.12% | 74.39% - 89.99% |

## H  COMPUTATIONAL COMPLEXITY

The computational complexity bound depends on the simulation of the inner loop (including simulation rounds $K$, aggregator set $\mathcal{A}$, and attack set $\mathcal{F}$) and problem dimensions of the outer loop (including number of clients $n$ and the dimension of gradients). We show the theoretical analysis

below and utilize empirical results to prove that it is worth to trade a little more computational complexity for a great robust model.

**Theoretical analysis** For RobustTailor with $K$ simulation rounds, our algorithm approaches a min-max equilibrium the best aggregation rule at the rate $O(K^{-1/2})$. In addition, the computational cost of RobustTailor is influenced by the server's aggregation rules and the adversary's attacks. If $n$ clients submit $d$-dimensional vectors, Krum's expected time complexity is $O(n^2 d)$ (Blanchard et al., 2017) and Comed's is $O(nd)$ (Pillutla et al., 2022).

For more fine-grained complexity analysis, suppose $\{\tilde{T}_1, \ldots, \tilde{T}_M\}$ denote the number of elementary operations to run each aggregation rule within the set of $M$ aggregation rules. The *worst-case* runtime complexity of RobustTailor per simulation round is determined by $\max_{i \in [M]} \tilde{T}_i$. However, the average complexity per round is the expected value of the number of elementary operations where the expectation is over the distribution of how likely each aggregator is chosen during simulation, which can be estimated empirically. Let us use $\tilde{p}_i$ to denote the probability of choosing $\mathcal{A}_i$. The average complexity per round is given by $\overline{T} = \sum_{i=1}^{M} \tilde{T}_i \tilde{p}_i$. Finally, the overall time complexity of RobustTailor is given by $O(\overline{T} K^{-1/2})$.

Moreover, the number of elementary operations for simulation can be much smaller than applying the actual aggregator on the model during training assuming the size of the public data is very small, which is typically the case in practice(Yoshida et al., 2020; Zhao et al., 2018). Note that our algorithm just adds computation complexity to the server while all clients remain the same cost based on their models and datasets. Therefore, it is worthwhile to trade slightly longer training time for a significantly improved training procedure w.r.t. robustness.

**Empirical results** We show computation costs and accuracy for different aggregation rules after running 15k iterations in reality, whose results are also shown in Fig. 1. We can see that RobustTailor still maintains a stable and high accuracy when facing a powerful adversary although it needs more computation time. However, Krum cannot reach a high accuracy and Comed shows a very unstable performance with lots of fluctuations when facing a strong adversary with AttackTailor. We note that compared with undesirable models of Krum and Comed, RobustTailor improves accuracy and stability drastically at the cost of slightly increased training time.

**Table 4:** Computational Complexity based on MNIST after running 15k iterations

| Aggregator | Time | Accuracy |
|---|---|---|
| Omniscient | 34 min | 96.63% |
| RobustTailor | 96 min | 85.87% |
| Krum | 37 min | 82.13% |
| Comed | 52 min | 90.74% |

**(a)** $\epsilon = 0.5$

| Aggregator | Time | Accuracy |
|---|---|---|
| Omniscient | 34 min | 96.63% |
| RobustTailor | 96 min | 92.03% |
| Krum | 37 min | 94.74% |
| Comed | 52 min | 60.39%-88.80% |

**(b)** $\epsilon = 100$

| Aggregator | Time | Accuracy |
|---|---|---|
| Omniscient | 34 min | 96.63% |
| RobustTailor | 190 min | 89.72% |
| Krum | 90 min | 80.98% |
| Comed | 121 min | 71.34%-89.75% |

**(c)** AttackTailor

