# OpenReview forum: "A Simulation-based Framework for Robust Federated Learning to Training-time Attacks"
_ICLR.cc/2023/Conference — Submitted to ICLR 2023_

### Official Review · Reviewer_GvuQ · 2022-10-24

**Confidence:** 4
**Correctness:** 3
**Technical Novelty And Significance:** 2
**Empirical Novelty And Significance:** 2
**Recommendation:** 5

**Clarity, Quality, Novelty And Reproducibility:**

Several important assumptions are not well justified.

First, it is assumed that the set of available attacks and the set of possible aggregators are common knowledge. In practice, however, it is more likely that the defender has limited knowledge about the attacks. An advanced attacker can either create a new attack or adapt an existing attack to make it unknown to the defender. Figures 9 and 10 in the appendix give some results in this direction. However, important details are missing. What is an attack "of the same type"? What is the "Mimic" attack? What is the "Alittle" attack?

Second, the paper assumes that each client donates a small subset of data to the server so that the server can simulate their behavior. Although this might be possible in certain application scenarios, obtaining a representative set of data samples can be challenging in general due to the heterogeneity among devices. Besides, malicious devices can share poisoned data samples with the server. Although Figure 11 in the appendix shows that the proposed method is still effective when each client only shares 0.1% of their local data, the simulation setting used to generate the figure is missing. Which dataset is used? Are the local data distributions iid or non-iid? What are the attacks and defenses considered?

Third, it is assumed the attacker has full knowledge of the model updates of all honest clients during training. Although this might be acceptable for earlier works on model poisoning (e.g., [Xie et al., 2020] and [Fang et al., 2020]) to demonstrate the worst-case damage that an attacker can possibly cause, it would be useful to consider more realistic attacks to understand the security of federated learning systems. Even in [Fang et al. 2020], more realistic partial knowledge attacks are developed in addition to full knowledge attacks.

Some implementation details are missing. What is the deep neural architecture used for CIFAR-10? How does the server generate the estimated \tilde{g}_i based on the public dataset?


**Strength And Weaknesses:**

The idea of switching defense policies dynamically to increase the attacker's uncertainty in adversarial machine learning has been considered before, although in a slightly different context [1]. The game theoretic model seems straightforward but reasonable. The main novelty of the paper is to simulate malicious devices' behavior using a small amount of honest data. However, the proposed method has some strong limitations.

First, from Algorithm 1, it is clear that the paper only considers non-adaptive attacks and defenses where a fixed rule is applied in each iteration. This can also be seen from the myopic loss function defined in (6). However, adaptive algorithms have been developed for both defenses [2,3] and attacks [4] in federated learning. Neither the algorithmic framework nor the analytic result in the paper applies to adaptive attacks or defenses.

Second, the paper ignores that most poisoning attacks and defenses have a set of adjustable hyperparameters that significantly affect their strength, making the possible attacks (and defenses) a very large or infinite set. The proposed solution only works for the finite case and suffers from a high complexity for large sets of attack and defense strategies. The experiments in Section 5 only consider a very small configuration with two defenses and two attacks with fixed parameters. Although more defenses are considered in Figure 8 in the appendix, the defense strategy is sampled from the pool uniformly at random without using the proposed solution.

Third, the game formulation ignores subsampling, a commonly used technique in large-scale federated learning. An advanced attacker will adapt its strategy according to the subset of malicious devices sampled in each iteration, creating another level of complexity unconsidered in the paper.

[1] Sengupta, Sailik, Tathagata Chakraborti, and Subbarao Kambhampati. Mtdeep: Moving target defense to boost the security of deep neural nets against adversarial attacks. International Conference on Decision and Game Theory for Security. 2019.

[2] Dan Alistarh, Zeyuan Allen-Zhu, and Jerry Li. Byzantine stochastic gradient descent. Advances in Neural Information Processing Systems(NeurIPS), 31, 2018

[3] Sai Praneeth Karimireddy, Lie He, and Martin Jaggi. Learning from history for byzantine robust optimization. In International Conference on Machine Learning(ICML), 2021

[4] Xiaoyu Cao, Minghong Fang, Jia Liu, and Neil Zhenqiang Gong. Fltrust: Byzantine-robust federated learning via trust bootstrapping. In NDSS, 2021



**Summary Of The Paper:**

This paper proposes a two-player zero-sum game for model poisoning attack and defense in federated learning. A powerful attacker with full knowledge of the FL system is considered, where the attacker controls a set of malicious devices that can share manipulated model updates with the server to lower the accuracy of the global model. In each FL iteration, the defender picks a defense rule randomly from a pool of robust aggregation-based defense algorithms by simulating the attacker's behavior using a small set of public data shared by honest devices. Similarly, the attacker picks an attack algorithm from a pool of attacks by simulating the defender's behavior using its knowledge of honest devices and the FL system. The paper formulates the attack-defender interaction in each FL iteration as a zero-sum game and adapts the Exp3 algorithm to solve the game on each side. Convergence analysis is provided together with a set of empirical results.

**Summary Of The Review:**

The paper proposes a game-theoretic framework to model poisoning attacks and defenses in federated learning where both the attacker and the defender adapt their behavior dynamically by simulating the behavior of the other side. Similar ideas have been considered in cybersecurity and adversarial learning. Although its application in federated learning seems new, the proposed method has strong limitations and relies on unrealistic assumptions.

---

> ### Author Response · Authors · 2022-11-11
> **Response to Reviewer GvuQ (Part 3)**
>
> ------
>
> - Adversary with partial knowledge.
>
> What we demonstrate is that $\mathrm{RobustTailor}$ still shows robustness even against a very strong adversary who has full knowledge of all honest clients. It implies robustness against a weaker adversary like an adversary just with partial knowledge (Fang et al. 2020).
>
> We agree it is important to consider more realistic attacks for a better understanding of the security of FL systems. Therefore, we show empirical results for the adversary with partial knowledge. For the partial knowledge setting, we assume that the adversary only knows the updates of two honest clients and designs compromised gradients based on them. We train a CNN model on MNIST under both iid and non-iid (the heterogenous degree $\mu=0.9$) settings (other experimental settings are the same as Section 5).  Numerical results are shown in the table below which compares with the full knowledge setting, and relevant figures are posted in [here](https://imgur.com/a/yTuyySq). The table below shows the results and compares them with the full knowledge setting. Most aggregation rules can perform at least the same as the scenario of the adversary with full knowledge. Note these results are also added in Appendix G.2 shown in Figure 15 and Table 4.
>
> 1. iid setting
>
> | Aggregator  | $\epsilon=0.5$ (Partial) |  $\epsilon=0.5$ (Full) || $\epsilon=100$ (Partial) |  $\epsilon=100$ (Full) || $\mathrm{AttackTailor}$ (Partial) |  $\mathrm{AttackTailor}$ (Full) |
> | ----  |  :-----: | :----: |-|  :-----: | :----: |-|  :-----: | :----: |
> |$\mathrm{RobustTailor}$|  88.84%  | 85.87%  ||  85.18% - 92.31% |  91.72%  ||  91.67%  |  90.54% |
> |Krum|           79.58%            | 82.13%  ||  94.74%   | 94.74%  ||  91.77%  |   81.68% |
> |Comed|  66.69% - 90.65%  | 72.07% - 90.74%  ||  69.83% - 89.32%   |  71.72% - 89.01% ||  67.13% - 90.06%  |  74.69% - 89.69% |
>
> 2. non-iid setting ($\mu=0.9$)
>
> | Aggregator  | $\epsilon=0.5$ (Partial) |  $\epsilon=0.5$ (Full) || $\epsilon=100$ (Partial) |  $\epsilon=100$ (Full) || $\mathrm{AttackTailor}$ (Partial) |  $\mathrm{AttackTailor}$ (Full) |
> | ----  |  :-----: | :----: |-|  :-----: | :----: |-|  :-----: | :----: |
> |$\mathrm{RobustTailor}$| 88.93%   | 87.54%  || 86.27% - 92.03%  |  90.62%  ||  91.51%  |  90.26% |
> |Krum|        73.50%       | 66.96%  ||  92.85%   | 92.85%  ||  90.60%  |   80.97% |
> |Comed|    80.85% - 90.54%     | 81.84% - 90.48%  ||  73.71% - 89.75%   |  75.73% - 89.38% ||  73.53% - 90.12%  |  74.39% - 89.99% |

---

> ### Author Response · Authors · 2022-11-11
> **Response to Reviewer GvuQ (Part 2)**
>
> ------
> - A typo in the description of Figure 8.
>
>
> RobustTailor with more defenses shown in Figure 8 is also based on simulation like Section 5. We thank the reviewer for pointing out the error and we have revised it in the paper.
>
>
> ------
> - How about an infinite set of aggregators or attacks due to hyperparameters, and the concerns about high complexity.
>
>
> It is indeed true that some aggregators and attacks include adjustable hyperparameters but it is not necessary for $\mathrm{RobustTailor}$ or $\mathrm{AttackTailor}$ to include all of them. We provide additional empirical results showing that just picking some representative hyperparameters is enough for ensuring a strong defense. We show the performance of $\mathrm{RobustTailor}$ when the actual attack configuration is outside of the server's used attack set in Appendix G.2. Specifically, Figure 9 shows that the true attack is the same type of attack as the server predicted, and it demonstrates that just having representative hyperparameters in $\mathrm{RobustTailor}$ suffice.
>
>
> ------
> - Provide description details in Figs. 9 and 10.
>
>
> We have added the details in the paper. To be specific, the attacks ($\epsilon=0.1$ and $\epsilon=150$) shown in Figure 9 belong to $\epsilon$-reverse attack, and they are the same type of attacks with $\epsilon=0.5/100$ used in the main text. The different types of attacks in Figure 10 are Mimic (Karimireddy et al., 2022) and Alittle (Baruch et al, 2019) attacks. To be specific, Mimic attack controls all Byzantine workers picking a good client to mimic and copy its output, while Alittle attack shifts the mean of the update leveraging on the large variance of the stochastic honest updates around the mean.
>
>
> ------
> - Provide implementation details of Figure 11.
>
>
> Except for the proportion of public data, other settings of 3 subfigures in Fig. 11 are the same as Fig. 1a, Fig. 1b, and Fig. 3 in the main text respectively. To be specific, we train a CNN model on MNIST dataset under the iid setting. For RobustTailor, Krum and Comed are in the server's defense setting; and for AttackTailor, $\epsilon=0.5$ and $\epsilon=100$ reverse attacks are considered.
>
>
> ------
>
> - Byzantine clients share poisoned data to the server.
>
> Sharing poisoned data samples with the server is a kind of data poisoning attack which has limited success to attack Byzantine-robust federated learning (Fang et al. 2020). Specifically, most existing aggregation methods like Krum and Comed can defend against data poisoning attacks successfully.
> To further confirm the minor impact on RobustTailor, we also show the empirical results below. Due to 16.7% of clients compromised by the adversary, we assume 16.7% of data in the public dataset is poisoned. We choose 2 normal data poisoning methods: 1) label flipping (LF) (Muñoz-González et al., 2017), and 2) random label (LR) (Zhang et al., 2021). Numerical results are shown in the table below and relevant figures are posted in [here](https://imgur.com/a/2JtBcG7). Poisoned data mixed in indeed influences $\mathrm{RobustTailor}$ especially facing a single attack but its performance is still stable and convincing. Note these results are also added in Appendix G.2 shown in Figure 14.
>
> | Aggregator  | $\epsilon=0.5$ | $\epsilon=100$ | $\mathrm{AttackTailor}$ |
> | :----:  |  :-----: | :----:| :----: |
> |$\mathrm{RobustTailor}$ (Origin)| 85.87% | 91.72%  | 90.54% |
> |$\mathrm{RobustTailor}$ (LF)| 87.31% | 91.72%  | 90.69%  |
> |$\mathrm{RobustTailor}$ (LR)| 87.11% | 88.16%  |  91.10%  |
> |  |  |  |
> |Krum| 82.31% | 91.72%  | 94.74%  |  81.68%  |
> |Comed| 72.07% - 90.74% | 71.72% - 89.01%  | 74.69% - 89.69%  |
>
>
> ------
> - Gap between public dataset and true samples.
>
> The relevant analysis of the difference between the public dataset and the underlying data distribution of honest clients is in Appendix A. It shows that the public dataset does not require to represent the true samples perfectly. In addition, the experiments above about poisoned samples also show that it is acceptable with such a gap which can be caused by the poisoned data from Byzantines or data heterogeneity.
>
>
>
> [Baruch et al, 2019] Gilad Baruch, Moran Baruch, and Yoav Goldberg. A little is enough: Circumventing defenses for distributed learning. In Proc. Advances in Neural Information Processing Systems (NeurIPS), 2019.
>
> [Muñoz-González et al., 2017] Luis Muñoz-González, Battista Biggio, Ambra Demontis, Andrea Paudice, Vasin Wongrassamee, Emil C Lupu, and Fabio Roli. Towards poisoning of deep learning algorithms with back-gradient optimization. In Proceedings of the 10th ACM workshop on artificial intelligence and security, 2017.
>
> [Zhang et al., 2021] Chiyuan Zhang, Samy Bengio, Moritz Hardt, Benjamin Recht, and Oriol Vinyals. Understanding deep learning (still) requires rethinking generalization. Communications of the ACM, 64(3):107–115, 2021.

---

> ### Author Response · Authors · 2022-11-11
> **Response to Reviewer GvuQ (Part 1)**
>
> We thank the reviewer for their thoughtful comments which we address one by one below.
>
>
> ------
> - The paper only considers non-adaptive attacks and defenses.
>
>
> Actually, we have considered adaptive aggregators and attacks. For adaptive aggregations, the centered clipping (CC) aggregator proposed by [3] is already present in $\mathrm{RobustTailor}$ for the experiments illustrated in Figure 8. Concerning the attacks, note that our proposed $\mathrm{AttackTailor}$ is an adaptive attack where the adversary is able to update the attack programs, their hyperparameters, and even the set of attack programs over time. It is indeed a general adaptive and simulation-based attack.
>
>
> We have cited [2,3] in related work. They are close works and both of them propose history-aided aggregators, and both of them can be added to $\mathrm{RobustTailor}$ framework. We already added CC of [3]. Algorithm 2 of [3] is a momentum-based method on the client side not an aggregation rule. Their time-varying approach is a modification of the **client-side** update rule to include momentum. It is thus compatible with our method, for which we implement it in basic $\mathrm{RobustTailor}$ (including Krum and Comed) and show the results in the table below. It seems that adding momentum slightly improves robustness, which is also reported in the literature (Ramezani-Kebrya et al., 2022).
>
>
> | Attack | With momentum | Without momentum |
> | ----  |  :-----: | :----: |
> |$\epsilon=0.5$|  87.80%  | 85.87%  |
> |$\epsilon=100$|  91.71%  | 91.72%  |
> |$\mathrm{AttackTailor}$|  90.93%  | 90.54%  |
>
>
> In addition, the adaptive attack in [4] is designed specifically for **linear aggregation rules**. However, many aggregators like Krum and Comed that we may consider for $\mathrm{RobustTailor}$ are non-linear. In this paper, we use $\mathrm{AttackTailor}$ which is a strong and adaptive attack designed to attack multiple aggregators with arbitrary structures.
>
>
> Finally, we want to emphasize that $\mathrm{RobustTailor}$ is a framework to boost existing aggregation rules rather than beat all of them. If a state-of-art aggregator comes out, it can be added to $\mathrm{RobustTailor}$ and improve the framework further.
>
>
> ------
> - Subsampling in FL.
>
> Subsampling is indeed important in federated learning. Therefore, we implement experiments to test it. We assume the server selects half of the clients asking for updates at each iteration which means that 6 of 12 clients are selected once in our implementation. The adversary can know which clients are selected and leverage the selected honest updates to design the attack for the compromised clients which are also chosen. As our assumption, the server knows the upper bound of the number of compromised clients $f$ and it still considers $f$ malicious updates each time. Other settings are the same as the basic setting in Section 5 except for the learning rate. We decrease the learning rate from 0.01 to 0.005 because all aggregation methods are too unstable under the original setting. The results are shown in [here](https://imgur.com/a/B2uQyem) which are also added in Figure 12 in Appendix G.2. Subsampling indeed decreases the stability of the system and increases the complexity of aggregation. In particular, Krum deals with subsampling well and $\mathrm{RobustTailor}$ performs acceptable but Comed is very unstable.
>
> Inspired by the suggestions of the reviewer, we propose that the adversary also can use a dynamic strategy that changes the number of malicious updates dynamically, which is similar to the subsampling of the server such that we run an additional experiment. All the settings are the same as in Section 5. However, the adversary picks 1-3 clients randomly to control at each iteration while the server still considers 2 Byzantines in 12 clients. The results are shown in [here](https://imgur.com/a/hENW65a). The table below shows the comparison with the origin result without the dynamic attack strategy. Overall, some aggregation rules would be impacted a little due to the increased complexity. But $\mathrm{RobustTailor}$ has a good performance which is consistent with the original results. Note these results are also added in Appendix G.2 shown in Figure 13 and Table 3.
>
> 1. $\epsilon=0.5$ reverse attack
>
> | Aggregator  | With dynamic attack | Without dynamic attack |
> | ----  |  :-----: | :----: |
> |$\mathrm{RobustTailor}$| 87.49% | 85.87%  |
> |Krum| 84.37% | 82.13%  |
> |Comed| 72.80% - 90.57% | 72.07% - 90.74%  |
>
> 2. $\epsilon=100$ reverse attack
>
> | Aggregator | With dynamic attack | Without dynamic attack |
> | ----  |  :-----: | :----: |
> |$\mathrm{RobustTailor}$| 85.26% - 91.62% | 91.72%  |
> |Krum| 93.88% | 94.74%  |
> |Comed| 71.80% - 88.15% | 71.72% - 89.01%  |
>
> 3. $\mathrm{AttackTailor}$ attack
>
> | Aggregator | With dynamic attack | Without dynamic attack |
> | ----  |  :-----: | :----: |
> |$\mathrm{RobustTailor}$| 90.17% |  90.54% |
> |Krum| 75.75% | 81.68%  |
> |Comed| 77.37% - 88.05% | 74.69% - 89.69%  |

---

> > ### Comment · Reviewer_GvuQ · 2022-11-17
> > **Further Comments**
> >
> > I'd like to thank the authors for providing a detailed response to my comments, which has resolved some of my concerns. Below are some further comments.
> >
> > 1. Although adaptive defenses can be incorporated, as shown in Figure 8, the proposed approach itself is non-adaptive. The no-regret learning algorithm and its analysis only apply to a single FL training iteration. Algorithm 2 itself does not use any historical information about attacks, and the loss function considers an immediate reward. The simulation results show that RobustTailor is more stable against uncertain attacks compared with Krum and Comed, both of which are non-adaptive. Figure 8 shows that RobustTailor+CC performs similarly to RobustTailor. However, I wonder how CC itself (with momentum) performs compared to RobustTailor.
> >
> > 2. Since RobustTailor uses a public dataset, I wonder how it compares with FLTrust [1] when a similar amount of public data is used. FLTrust obtains convincing performance against several attacks using a very small public dataset (100 images for MNIST).
> >
> > 3. The paper considers a very small FL setting with 12 devices including 2 malicious devices. This is far from being realistic. I would suggest considering a setting with at least 100 devices and 10-20% subsampling.
> >
> > [1] Xiaoyu Cao, Minghong Fang, Jia Liu, Neil Zhenqiang Gong. FLTrust: Byzantine-robust Federated Learning via Trust Bootstrapping. NDSS 2021.

---

> > > ### Author Response · Authors · 2022-11-19
> > > **Further response to Reviewer GvuQ**
> > >
> > > We thank the reviewer for the quick reply. We address the concerns one by one below.
> > >
> > > ------
> > > - Results of CC and FLTrust.
> > >
> > > We show the results of CC and FLTrust in the tables below. Comparing $\mathrm{RobustTailor}$  including only Krum and Comed with CC/FLTrust is unfair so we also show the results of $\mathrm{RobustTailor}$ including Krum, Comed and CC/FLTrust. CC has good performance under $\epsilon$-reverse attacks (it is vulnerable to "A little"-type attacks), and $\mathrm{RobustTailor}$ gives not the best but convincing performance. Recently, (Ozfatura et al., Figure 5) shows that CC is vulnerable to ROP attack while TM is robust to such attack. We note that $\mathrm{RobustTailor}$ including TM can defend ROP attack. This also proves that no aggregator defending all attacks successfully is a reasonable and realistic assumption.
> > >
> > > | Aggregator  | $\epsilon=0.5$ | $\epsilon=100$ | $\mathrm{RobustTailor}$ |
> > > | :----  |  :-----: | :----:| :----:|
> > > | CC | 92.88% | 94.23%  |  93.47%  |
> > > |$\mathrm{RobustTailor}$ (Krum, Comed)| 85.87% | 91.72%  |   90.54%  |
> > > |$\mathrm{RobustTailor}$ (Krum, Comed, CC) | 86.85% | 91.74%  | 91.13% |
> > >
> > >
> > > | Aggregator  | $\epsilon=0.5$ | $\epsilon=100$ | $\mathrm{RobustTailor}$ |
> > > | :----  |  :-----: | :----:| :----:|
> > > | FLTrust | 88.39% | 88.39%  | 88.39%  |
> > > |$\mathrm{RobustTailor}$ (Krum, Comed)| 85.87% | 91.72%  |    90.54%  |
> > > |$\mathrm{RobustTailor}$ (Krum, Comed, FLTrust) | 85.02% | 92.30%  | 90.32% |
> > >
> > > Importantly, we want to emphasize that $\mathrm{RobustTailor}$ is not an aggregation rule based on Krum and Comed. It is a **framework** which can pick a reliable aggregator from a set of existing aggregation rules at each iteration. Any aggregation rules can be included in $\mathrm{RobustTailor}$ even those that will be proposed in the future.
> > >
> > > Why do we choose Krum and Comed? We want to deal with a situation that both the server and the adversary have a set of aggregators and attacks respectively and no aggregator or attack is omnipotent, which is close to the reality since it is easy for an informed adversary to tailor training-time attacks against a specific aggregator (Fang et al., 2020). We focused on Krum and Comed which are vulnerable to $\epsilon=0.5$ and $\epsilon=100$ separately. The current results demonstrate that $\mathrm{RobustTailor}$ indeed chooses the optimal one from its set and promises a reliable performance. However in realistic implementation, you can choose other aggregators e.g., CC/FLTrust/TM.
> > >
> > > ------
> > > - Subsampling with more clients.
> > >
> > > We run additional experiments with 120 clients, in which 20 clients are Byzantines. The server chooses 10% of clients randomly for aggregation at each iteration, and it assumes that 2 of 12 clients are compromised every iteration. Note that other settings of experiments are the same as Figure 1 (a), Figure 1 (b), and Figure 3 in the paper except for 0.001 learning rate for training. The results are shown in the table below and relevant figures are posted in [here](https://imgur.com/a/d98010S), which are also updated in Appendix G.2 shown in Figure 11. The results of subsampling with 120 clients are consistent with the subsampling with 12 clients we posted before. $\mathrm{RobustTailor}$ still shows convincing results.
> > >
> > > | Aggregator  | $\epsilon=0.5$ | $\epsilon=100$ | $\mathrm{AttackTailor}$ |
> > > | :---- |  :-----: | :----:| :----: |
> > > |Omniscient| 91.41% | 91.41%  | 91.41% |
> > > |$\mathrm{RobustTailor}$| 89.06% | 85.59%  | 87.17% |
> > > |Krum| 73.03% | 86.79%  | 74.59%  |
> > > |Comed| 89.28% | 82.87%  | 86.33%  |
> > >
> > > In addition, our implementation is not far from realistic. In cross-silo FL, the number of clients is usually small (Kairouz et al., 2021). Our implementation can be used in such situations like the cooperation of banks or hospitals.
> > >
> > > [Ozfatura et al., 2022] Kerem Ozfatura, Emre Ozfatura, Alptekin Kupcu, and Deniz Gunduz. Byzantines can also learn from history: Fall of centered clipping in federated learning. arXiv preprint arXiv:2208.09894, 2022.

---

> ### Author Response · Authors · 2022-11-16
> **Follow-up**
>
> Dear Reviewer GvuQ,
>
> We are currently unaware of the extent to which our responses have clarified your concerns, and willing to provide further clarifications to you.
>
> Regards,
>
> Authors

---

> ### Author Response · Authors · 2022-12-06
> **Happy to provide further clarification**
>
> Dear Reviewer GvuQ,
>
> We hope that we have resolved the major concerns from your side. In particular, regarding "comparison with CC and FLTrust", we hope that our additional experiments and discussion clarified your concerns. If possible, could you please rate the paper more positively? We would be happy to provide further clarifications if needed.
>
> Regards,
>
> Authors

---

### Official Review · Reviewer_ovJ3 · 2022-10-25

**Confidence:** 5
**Clarity, Quality, Novelty And Reproducibility:** The paper is technically interesting.
**Correctness:** 4
**Technical Novelty And Significance:** 4
**Empirical Novelty And Significance:** 4
**Recommendation:** 5

**Strength And Weaknesses:**

Strength:

(1) This paper captured a very interesting observation in game theory that the averaged historical policy produced by no-regret learners can approximate a Nash Equilibrium when the game is played repeatedly. This is a very cute idea that naturally matches the federated learning problem in presence of Byzantine attacks. The methodology is also novel in the federated learning domain.

(2). The paper provided strong and extensive empirical results and demonstrated that the proposed defense indeed enhances the robustness of federated learning against training-time attacks.

Weaknesses:

(1). Most theoretical results in this paper are not novel. In particular, The Lemma 1 and 2 are both existing results in the game theory and multi-armed bandit domain. The Lemma 3 is somewhat novel, but it seems to be an easy application of Lemma 1 and 2. Therefore, while the paper provided theoretical guarantees, the results are not surprising or novel enough to me. Besides that, I think in a lot of places, the paper missed a min operator, e.g., in the regret definition (8) and also (10). Please add the min operator to avoid confusion.

(2). The paper relied on a public dataset to construct some simulator, and then uses the simulator to obtain estimated loss/reward for the aggregator and attacker. Therefore, the regret that Algorithm 2 tries to minimize is actually the objective on the public dataset, instead of the ground-truth environment. Although the authors provided several papers to support this assumption of having a simulator, I am kind of
curious what if there is a distribution gap between the public dataset and the global environment. It would be interesting to discuss this problem and provide some empirical study.

(3). I believe the authors should consider using EXP3.P instead of EXP3 to do experiments. This is because although EXP3 achieve no regret, it requires an important condition that the loss functions over time are pre-determined before the game starts. However, in this paper, the attacker can be adaptive to the behavior of the aggregator over time. That means the policy q_t selected by the adversary can depend on the historical p_t selected by the aggregator. As a result, the loss function for the aggregator at time t is p*E(L)*q_t^T also depends on the historical p's (i.e., p_1, ..., p_{t-1}), and thus is not determined beforehand. The EXP3 does not provide strict theoretical guarantee in this case. Instead, the EXP3.P would work.

**Summary Of The Paper:**

This paper proposed a minimax formulation to model the attacks and defenses in federated machine learning. The aggregator is an agent who wants to maximize the accuracy in presence of Byzantine clients, where the Byzantine clients want to corrupt the performance of the aggregated model. Therefore, federated learning with Byzantine attacks can be viewed as a two-player zero-sum game. To estimate the loss value for the two players, the paper proposed using public dataset to construct a simulator of the environment. To further reduce the computational time, the authors considered using bandit feedback, in which only the loss for the selected aggregator and tailored attack is observed. Fortunately, by using no-regret bandit algorithms, the averaged historical policy converges to some Nash Equilibrium, and thus the output aggregator selection policy is optimal in the sense of NE. The paper provided extensive experimental results and demonstrate that the proposed defense can indeed improve the robustness of federated learning against Byzantine attacks.

**Summary Of The Review:**

I worked in related areas.

---

> ### Author Response · Authors · 2022-11-14
> **Response to Reviewer ovJ3**
>
> We thank the reviewer for their thoughtful comments which we address one by one below.
>
> ------
> - Theoretical results are not novel.
>
> We agree that this kind of result is most likely stated elsewhere in the literature, but we have not been able to recover the exact form we need, which is why we rederive it in the paper. We have clarified that this type of result is well-known in the community by explicitly stating that Lemma 1 is folklore.
>
> ------
> - Clarify the missing of a min operator.
>
> Notice that we state that (8) and (10) holds for all $i,j$. This formulation is equivalent to assuming the conditions hold for the best choice of $i,j$. That is, if the condition holds for all choices it also holds for the optimal choice and conversely if it holds for the optimal choice the inequality must also hold for any other (worse) choice. This way of expressing regret can be found elsewhere in the literature, see for example section 1.2 of (Cesa-Bianchi et al., 2021). We favour this notation since it simplifies some of the steps in the reasoning, but are willing to change if this leads to misunderstandings.
>
>
> ------
> - Gap between public dataset and true samples.
>
> The difference between the public dataset and the underlying data distribution of honest clients is analyzed in Appendix A. It shows that the public dataset doesn't be required to represent the true samples perfectly. We also agree that it is more convincing to do some empirical studies. Based on our model, we assume $f$ compromised clients share poisoned data to the public dataset.
>
> Due to 16.7% of clients compromised by the adversary in our experimental setting, we assume 16.7% of data in the public dataset is poisoned. We choose 2 normal data poisoning methods: 1) label flipping (LF) (Muñoz-González et al., 2017), and 2) random label (LR) (Zhang et al., 2021). Numerical results are shown in the table below and relevant figures are posted in [here](https://imgur.com/a/2JtBcG7). Poisoned data mixed in indeed influences $\mathrm{RobustTailor}$ especially facing a single attack but its performance is still stable and convincing. Note these results are also added in Appendix G.2 shown in Figure 14.
>
> | Aggregator  | $\epsilon=0.5$ | $\epsilon=100$ | $\mathrm{AttackTailor}$ |
> | :----:  |  :-----: | :----:| :----: |
> |$\mathrm{RobustTailor}$ (Origin)| 85.87% | 91.72%  | 90.54% |
> |$\mathrm{RobustTailor}$ (LF)| 87.31% | 91.72%  | 90.69%  |
> |$\mathrm{RobustTailor}$ (LR)| 87.11% | 88.16%  |  91.10%  |
> |  |  |  |
> |Krum| 82.31% | 91.72%  | 94.74%  |  81.68%  |
> |Comed| 72.07% - 90.74% | 71.72% - 89.01%  | 74.69% - 89.69%  |
>
>
> ------
> - Implement EXP3.P rather than EXP3.
>
> We additionally test EXP3.P as requested. The experimental results can be found [here](https://imgur.com/a/HyhlOmy) under three different settings. The comparison shows that the method has very similar performance to EXP3. We remain at the disposal of the reviewer for any further questions.
>
>
>
> [Cesa-Bianchi et al., 2021] Nicolo Cesa-Bianchi and Francesco Orabona. Online learning algorithms. Annual review of statistics and its application, 2021.
>
> [Muñoz-González et al., 2017] Luis Muñoz-González, Battista Biggio, Ambra Demontis, Andrea Paudice, Vasin Wongrassamee, Emil C Lupu, and Fabio Roli. Towards poisoning of deep learning algorithms with back-gradient optimization. In Proceedings of the 10th ACM workshop on artificial intelligence and security, 2017.
>
> [Zhang et al., 2021] Chiyuan Zhang, Samy Bengio, Moritz Hardt, Benjamin Recht, and Oriol Vinyals. Understanding deep learning (still) requires rethinking generalization. Communications of the ACM, 64(3):107–115, 2021.

---

> ### Author Response · Authors · 2022-11-17
> **Follow-up**
>
> Dear Reviewer ovJ3,
>
> We are currently unaware of the extent to which our responses have clarified your concerns, and willing to provide further clarifications to you.
>
> Regards,
>
> Authors

---

> ### Author Response · Authors · 2022-12-06
> **Happy to provide further clarification**
>
> Dear Reviewer ovJ3,
>
> We hope that we have resolved the major concerns from your side. In particular, regarding "gap between public dataset and true samples", we hope that our additional experiments and discussion clarified your concerns. If possible, could you please rate the paper more positively? We would be happy to provide further clarifications if needed.
>
> Regards,
>
> Authors

---

### Official Review · Reviewer_rcbh · 2022-10-25

**Confidence:** 4
**Correctness:** 4
**Technical Novelty And Significance:** 2
**Empirical Novelty And Significance:** 2
**Recommendation:** 5

**Clarity, Quality, Novelty And Reproducibility:**

The presentation is clear, and results look sound. The idea of formulating robust learning as a game against an adversary may be novel in the specific literature on federated learning.

**Strength And Weaknesses:**

Strength: The paper studies an interesting and well-motivated problem and takes effort to model it and formulate it as a game. The paper is clear and easy to follow and results look complete.

Weakness: Theoretical contribution is a bit thin. The model looks a bit simplistic and hence more like one "on paper". Some assumptions are a bit too strong. For example, the assumption that both players choose their algorithms from finite sets known to each other is a bit too strong, and seems more for the sake of formulating the problem as a normal-form game.

**Summary Of The Paper:**

The paper introduces a game theoretic model to study defense against adversarial attacks in federated learning. The task of designing a robust learning algorithm against potential attacks is formulated as the problem of computing a good strategy in a game. The authors presented bandit algorithms to compute the equilibrium of the game as well as several results about the theoretical guarantee of the algorithm. Experiments were also conducted to evaluate the performance fo the algorithm and the quality of solutions it generates.

**Summary Of The Review:**

A well-presented paper on a well-motivated problem, with some novelty in the specific literature. Results look sound but a bit thin on the theoretical side.

---

> ### Author Response · Authors · 2022-11-11
> **Response to Reviewer rcbh**
>
> We thank the reviewer for their thoughtful comments which we address one by one below.
>
> ------
>
> - Simplicity of the formulation:
>
> We agree that the approach might be considered simple (at least in hindsight), but the numerical results importantly demonstrate that such an approach can stabilize training. This meta-level of aggregation and attack selection from a family of choices seems to have been largely unexplored in the literature. In that context, this work can be seen as a first step which will hopefully bring awareness to this design space. If the reviewer has any specific suggestions in mind we would be happy to discuss and include them in the revision.
>
> ------
>
> - Clarify the finite set of aggregators or attacks:
>
> One reason the reviewer might think a finite set of aggregators and attacks is a strong assumption is that some attacks might be unknown to the server and vice versa. We note that $\mathrm{RobustTailor}$ as an adaptive strategy mixing several aggregators is naturally more robust than the individual underlying aggregators. To demonstrate this property, we attack the server with an attack type outside the set of known attacks (see Figure 10). We observe that $\mathrm{RobustTailor}$ still performs well. Therefore, even just with a finite set of known attacks, $\mathrm{RobustTailor}$ can still improve the robustness.
>
> Another reason the assumptions might be considered strong may be that some aggregators and attacks include adjustable hyperparameters. We provide additional empirical results showing that just picking some representative hyperparameters is enough for ensuring a strong defense. We show the performance of $\mathrm{RobustTailor}$ when the actual attack configuration is outside of the server's used attack set in Appendix G.2. Specifically, Figure 9 shows that the true attack is the same type of attack as the server predicted, and it demonstrates that just having representative hyperparameters in $\mathrm{RobustTailor}$ suffice.

---

> ### Author Response · Authors · 2022-11-16
> **Follow-up**
>
> Dear Reviewer rcbh,
>
> We are currently unaware of the extent to which our responses have clarified your concerns, and willing to provide further clarifications to you.
>
> Regards,
>
> Authors

---

> ### Author Response · Authors · 2022-12-06
> **Happy to provide further clarification**
>
> Dear Reviewer rcbh,
>
> We hope that we have resolved the major concerns from your side. In particular, regarding "finite set of aggregators", we hope that our explanation of attacking outside the set of known attacks clarified your concerns. If possible, could you please rate the paper more positively? We would be happy to provide further clarifications if needed.
>
> Regards,
>
> Authors

---

### Official Review · Reviewer_vXe3 · 2022-10-26

**Confidence:** 4
**Correctness:** 3
**Technical Novelty And Significance:** 2
**Empirical Novelty And Significance:** 2
**Recommendation:** 5

**Clarity, Quality, Novelty And Reproducibility:**

The paper is clear to read.
I think the idea is novel, but I have a some basic questions - see weaknesses.


**Strength And Weaknesses:**

I am surprised that no-regret algorithm converging to NE in a two player zero-sum game is presented as a new result. This is a very well-known result, and appears in lecture notes and tutorials also, e.g., see page 33 in https://algo.cs.uni-frankfurt.de/lehre/agt/winter1920/folien/correlated.pdf

I believe a main purpose of federated learning is to preserve privacy of private data-sets. The assumption of the server having access to a dataset from every client (even one that mimics client data distribution but is not the actual data) is a loss of privacy. The authors have cited a number of previous works to support this (sorry, I have not read these cited papers), but have not really discussed why this kind of access to dataset is fine from a privacy perspective. Please provide a detailed response to this question.

I am bit lost in notations, but is there some assumption about \tilde{g} such as unbiased estimate? Along similar lines, isnt there some assumption about the simulation that could relate Sim-MinMax to MinMax. Solving Sim-MinMax is fine only if that is close (close in some sense that must be defined) to MinMax.

Minor point: the footnote on page 1 refers to an experiment in the appendix, if this point was important to mention in Introduction, maybe this experiment should be in main paper.


**Summary Of The Paper:**

The authors propose an attack on federated learning where a subset f clients are compromised. The problem is formulated as a game where both players use a no-regret learning algorothm to converge to the NE. The game is simulated, as the defender does not know the true updates of compromised clients.

**Summary Of The Review:**

A good overall idea, but I am not sure of claims made in the paper being novel or interesting.

---

> ### Author Response · Authors · 2022-11-14
> **Response to Reviewer vXe3**
>
> We thank the reviewer for their thoughtful comments which we address one by one below.
>
> ------
> - No-regret results for a two-player zero-sum game are known.
>
> We suspect the reviewer is referring to the slide named “No-Regret and Optimal Strategies” [here](https://algo.cs.uni-frankfurt.de/lehre/agt/winter1920/folien/correlated.pdf). This result is not exactly what we need. We are not interested in bounding the given sequence of the opponents play ($L_H^T$ in the notation of the slides) but rather the loss given the optimal play of the opponent. This is why we get a dependency on the regret of the opposing player as well (which the slides do not have). We agree that this kind of result is most likely stated elsewhere in the literature, but we have not been able to recover it, which is why we rederive it in the paper. We have clarified that this type of result is well-known in the community by explicitly stating that Lemma 1 is folklore.
>
>
> ------
> - Elaborate on using the public dataset from the perspective of privacy.
>
> In Remark 1, we explained that providing such a public dataset is a common assumption in FL. It is a price for clients to become robust against training-time attacks. Here are some reasons. First, the amount of the public dataset is small. Figure 11 demonstrates that even 0.1% of data from clients can promise the performance of $\mathrm{RobustTailor}$. Second, the public dataset can be from some clients' data that is not private-sensitive or from a publicly available data source (Kairouz et al., 2021). Third, to protect the raw data, clients can submit the distillation of the raw data (Wang et al., 2018) to protect privacy. Fourth, it’s acceptable that there is a small gap between the public dataset and the true data distribution. An additional experiment that poisoned data mixed in the public dataset shown in Figure 14 demonstrates that such a small gap does not reduce the effectiveness of RobustTailor substantially. Finally, we note that many aggregation rules use auxiliary data. For example,  Fang et al. (2020), Xie et al. (2020), Cao et al. (2019), and Cao et al. (2020) propose server-side verification methods using auxiliary data, in which the server asks for a small clean dataset from their clients. Therefore, $\mathrm{{RobustTailor}}$ has good expandability that these aggregators can be added to its pool.
>
> Beyond machine learning, economists have observed a privacy paradox where while people claim that they care about privacy, they are willing to relinquish private data quite easily when incentivized to do so (Athey et al., 2017; Acemoglu et al., 2022). We think additional robustness provides such incentives.
>
>
> ------
> - Clarify $\tilde{g}$ and relate Sim-MinMax to MinMax.
>
> The estimate of honest update $\tilde{g}$ is a **rough estimate** of the ideal $g^*$ mentioned in Remark 1. Appendix B also discusses the requirements relating to the estimated gradient $\tilde{g}$, the true gradient $\nabla F(x)$, and the output of aggregation.
>
> Also in Appendix B, the conditions for relating Sim-MinMax to MinMax are given. Two assumptions for the almost sure convergence of $\mathrm{RobustTailor}$ are 1) the public data donated by clients is representative of the underlying data distribution of honest clients, and 2) the number of Byzantine clients is sufficiently small ($n \geq 2f+1$).
>
>
> ------
> - Footnote on page 1.
>
> Thanks for this suggestion. It is hard to decide which experiments should be shown in the main text due to the page limitation. We have instead pointed to the explicit section in the supplementary so it is easier for the reader to find the relevant experiments.
>
>
> [Athey et al., 2017] Susan Athey, Christian Catalini, and Catherine Tucker. The digital privacy paradox: Small money, small costs, small talk. Technical report, National Bureau of Economic Research, 2017.
>
> [Acemoglu et al., 2022] Daron Acemoglu, Ali Makhdoumi, Azarakhsh Malekian, and Asu Ozdaglar. "Too much data: Prices and inefficiencies in data markets." American Economic Journal: Microeconomics 14, no. 4 (2022): 218-56.

---

> ### Author Response · Authors · 2022-11-17
> **Follow-up**
>
> Dear Reviewer vXe3,
>
> We are currently unaware of the extent to which our responses have clarified your concerns, and willing to provide further clarifications to you.
>
> Regards,
>
> Authors

---

> > ### Comment · Reviewer_vXe3 · 2022-11-17
> > **Still some concerns**
> >
> > Thank you for the response. I am still concerned about no regret convergence - I am sure the authors can pull up a reliable source (instead of folklore). Also, that lecture notes slide 30-34 is where the bound can also be obtained, as an easy corollary at best.
> >
> >
> > I could not find Figure 14 in the write-up? There are figures only till Figure 13.
> >
> >
> > Also, I understand that Appendix B has some claims about closeness of \tilde{g} but this appendix is written quite non-rigorously. As in, where is g^* and what are small f_i (I thought we had capital F_i in main paper).

---

> > > ### Author Response · Authors · 2022-11-18
> > > **Further response to Reviewer vXe3**
> > >
> > > We thank the reviewer for the reply, and we address the concerns one by one below.
> > >
> > > ------
> > > - Clarify folklore.
> > >
> > > The best source we were able to find so far on this kind of result is the lecture notes of Dugmi 2017. We have now clarified immediately after Lemma 1. If the reviewer is aware of a more appropriate source we kindly ask to let us know.
> > >
> > > Shaddin Dughmi, Omkar Thakoor, and Umang Gupta. CSCI699: [Topics in learning & game
> > > theory: Lecture 6](https://viterbi-web.usc.edu/~shaddin/cs699fa17/lectures/lec6.pdf). pp. 11, 2017.
> > >
> > > ------
> > > - Clarify the figure of poisoned data mixed in.
> > >
> > > The figure naming has changed since the first revision due to the latest revision that restructured the paper. The figure named “Poisoned data mixed in the public dataset” is shown in Figure 4 in Section 5 now. We apologize for the inconvenience.
> > >
> > > ------
> > > - Appendix B.
> > >
> > > $\mathbf{g}^*$ is defined in equation (1), which is the empirical mean of all honest updates without an attack. The lowercase $f_i$ in Appendix B was a typo and we have revised it to $F_i$ in the paper.

---

> ### Author Response · Authors · 2022-12-06
> **Happy to provide further clarification**
>
> Dear Reviewer vXe3,
>
> We hope that we have resolved the major concerns from your side. In particular, regarding "no regret convergence", we hope that our revision clarified your concerns. If possible, could you please rate the paper more positively? We would be happy to provide further clarifications if needed.
>
> Regards,
>
> Authors

---

### Author Response · Authors · 2022-11-17
**New Revision Uploaded**

Dear reviewers,

We uploaded the new revision of our paper. Section 5 EXPERIMENTAL EVALUATION is reconstructed. Specifically, we moved the results of 'poisoned data mixed in the public dataset' and 'unknown attacks for the server' to the main body. The remaining additional experiments can still be found in Appendix G.2: 1) subsampling by the server; 2) dynamic strategy of the adversary; 3) adversary with partial knowledge. We are looking forward to your further suggestions.

Regards,

Authors

---

### Decision · Program_Chairs · 2023-01-20

**Decision:**

Reject

**Justification For Why Not Higher Score:**

The analysis is not novel (enough), reviewers still have concerns on the privacy matter for sharing a sample of the data; the experiments are not thorough enough.

**Justification For Why Not Lower Score:**

N/A

**Metareview: Summary, Strengths And Weaknesses:**

In this paper, the authors studied a gam-theoretical formulation to model attacks/defenses in the setting of federated learning. The reviewers anonymously agree that the problem is important, the idea is interesting, and the the paper is in general well-written. On the  other hand, all reviewers have various reservations on accepting this paper, even after the rebuttal. The main concerns include: the theoretical analysis is not very novel; the assumption of sharing a (subset) of the dataset to the public hurts privacy; the experiments are not thorough enough as a security paper. After reading the paper myself, I agree with reviewers that the paper in the current form (even the revised version) is not ready to be accepted. The reviewers have provided thorough comments which I strongly encourage the authors to take serious consideration when preparing for the next submission.